



# Does a difference in ice sheets between Marine Isotope Stages 3 and 5a affect the duration of stadials?

Sam Sherriff-Tadano[1, 6], Ayako Abe-Ouchi[1, 2, 3], Akira Oka[1], Takahito Mitsui[4, 5], Fuyuki Saito[2]

[1]Atmosphere and Ocean Research Institute, The University of Tokyo, Kashiwa, Japan
[2]Japan Agency for Marine-Earth Science and Technology, Yokohama, Japan
[3]National Institute of Polar Research, Tokyo, Japan
[4]Department of Mathematics and Computer Science, Free University of Berlin, Berlin, Germany
[5]Potsdam Institute for Climate Impact Research, Potsdam, Germany
[6]School of Earth and Environment, University of Leeds, Leeds, United Kingdom

*Correspondence to*: Sam Sherriff-Tadano (S.Sherriff-Tadano@leeds.ac.uk)

**Abstract.** Glacial periods undergo frequent climate shifts between warm interstadials and cold stadials on a millennial time-scale. Recent studies have shown that the duration of these climate modes varies with the background climate; a colder background climate and lower $CO_2$ generally results in a shorter interstadial and a longer stadial through its impact on the Atlantic Meridional Overturning Circulation (AMOC). However, the duration of stadials was shorter during the Marine Isotope Stage 3 (MIS3) compared with MIS5, despite the colder climate in MIS3, suggesting potential control from other climate factors on the duration of stadials. In this study, we investigated the role of glacial ice sheets. For this purpose, freshwater hosing experiments were conducted with an atmosphere–ocean general circulation model under MIS5a, MIS3 and MIS3 with MIS5a ice sheet conditions. The impact of ice sheet differences on the duration of the stadials was evaluated by comparing recovery times of the AMOC after freshwater forcing was reduced. Hosing experiments showed a slightly shorter recovery time of the AMOC in MIS3 compared with MIS5a, which was consistent with ice core data. We found that larger glacial ice sheets in MIS3 shortened the recovery time. Sensitivity experiments showed that stronger surface winds over the North Atlantic shortened the recovery time by increasing the surface salinity and decreasing the sea ice amount in the deepwater formation region, which set favourable conditions for oceanic convection. In contrast, we also found that surface cooling by larger ice sheets tended to increase the recovery time of the AMOC by increasing the sea ice thickness over the deepwater formation region. Thus, this study suggests that the larger ice sheet in MIS3 compared with MIS5a could have contributed to the shortening of stadials in MIS3, despite the climate being colder than that of MIS5a, when the effect of surface wind played a larger role.



## 1 Introduction

Reconstructions from ice cores reveal that the climate varied frequently on a millennial time-scale over the glacial period

(Kawamura et al. 2017). These millennial-scale climate variabilities are known as Dansgaard–Oeschger (DO) cycles, and occurred more than 20 times over the last glacial period (DO cycles, Fig. 1, Dansgaard et al. 1993, Huber et al. 2006, Capron et al. 2010, Kindler et al. 2014). The DO cycles are famous for their abrupt and large temperature increases over Greenland from stadial to interstadial, followed by gradual cooling and a drastic return to the stadial conditions. These two contrasting climate modes persisted for more than several hundred years, and in total, resulted in periodicity from one thousand years to

more than five thousand years (Buizert and Schmittner 2015, Kawamura et al. 2017). The DO cycles are often attributed to reorganizations of the Atlantic meridional overturning circulation (AMOC) between a vigorous mode and a weak mode (Ganoploski and Rahmstrof 2001, Piotrowski et al. 2005, Menviel et al. 2014, Henry et al. 2016, Menviel et al. 2020). For example, it has been shown that the shift of the AMOC from a vigorous mode to a weak mode caused a reduction of northward oceanic heat transport in the Atlantic, expansion of sea ice and drastic cooling over the North Atlantic and warming over the

Southern Ocean (Kageyama et al. 2010, 2013).

To better understand the dynamics of DO cycles as well as the spread in the duration of DO cycles, previous studies investigated possible relations between the frequency of these cycles and the background climate such as glacial ice sheet amounts and atmospheric $CO_2$. For example, McManus et al. (1999) suggested that DO cycles occurred most frequently when the size of

the glacial ice sheets was at an intermediate level between interglacial and full glacial. They suggested that intermediate ice sheets could be unstable, and that the frequent release of freshwater could cause drastic weakening of the AMOC. On the other hand, ice core and modelling studies have suggested the importance of global cooling in determining the frequency of DO cycles (Buizert and Schmittner 2015, Kawamura et al. 2017). Kawamura et al. (2017) showed that DO cycles occurred most frequently when the Antarctic temperature and global cooling were at intermediate levels between interglacial and full glacial

periods over the last 720 thousand years. It was further demonstrated based on climate modelling experiments that the vigorous AMOC becomes more vulnerable to perturbations such as freshwater hosing when the global or Southern Ocean climate is colder than the modern climate but not as cold as the full glacial climate, resulting in a more unstable vigorous AMOC mode during mid-glacial periods (Buizert and Schmittner 2015, Kawamura et al. 2017). These results suggest that the spread of the frequency of DO cycles may not purely result from chaotic behaviour of the AMOC, but rather may be modulated by changes

in the background climate (Buizert and Schmittner 2015, Kawamura et al. 2017, Mitsui and Crucifix 2017).

Recent studies of ice cores from both Greenland and Antarctica further explored the relation of the background climate and the frequency of DO cycles by separating the durations of interstadials and stadials. With respect to interstadials, Buizert and Schmittner (2015) showed that the duration decreased as the Antarctic temperature decreased from interglacial to full glacial

conditions (Fig. 1). Lohmann and Ditlevsen (2019) also showed, based on ice core data from Greenland, that the duration of



interstadials was highly correlated with the surface cooling rate over the northern North Atlantic; the duration decreased as the cooling rate of the Greenland temperature increased. These studies are supported by experiments with climate models showing an increased sensitivity of the vigorous AMOC to freshwater hosing under colder climates (Zhang et al. 2014b, Kawamura et al. 2017), and by climate model studies showing shortening of the duration of interstadials in their intrinsic millennial-scale

climate variability with lower $CO_2$ levels (Brown and Galbraith 2016, Klockmann et al. 2018).

With respect to stadials, the situation is different. Buizert and Schmittner (2015) found a weak relation between the durations of stadials and Antarctic temperature; the durations of the stadials were extremely long during the full glacial interval (MIS2, 4, Fig. 1a), short in the early glacial interval (Marine Isotope Stage 5 (MIS5)), and even shorter in the mid-glacial period

(MIS3, Fig. 1b), which contributed to the short periodicity of DO cycles during mid-glacial periods. In addition, Lohmann (2019) analyzed the dust record in Greenland ice cores and found that the durations of stadials correlated with the decreasing trend of dust during the first 100 years of the stadials. Although the factors controlling the trend of dust remain unclear, these results suggest that another type of climate forcing over the North Atlantic played a role in modulating the durations of stadials in combination with surface cooling. In addition, these results suggest that the processes modulating durations of interstadials

and stadials may differ. Nevertheless, it still remains unclear why the durations of stadials were generally shorter in MIS3 compared with MIS5, despite colder conditions in MIS3.

From a climate modelling point of view, previous studies have shown that the recovery time of the AMOC and the duration of stadials depend on the background climate, based on freshwater hosing experiments conducted under different background

climate conditions. For example, Weber and Drijhout (2007) showed in simulations with an earth system model of intermediate complexity (EMIC) that the recovery time of the AMOC was longer under glacial conditions (Last Glacial Maximum, LGM) compared with preindustrial (PI) conditions. Bitz et al. (2007) also showed with a comprehensive climate model that the recovery time became longer under the LGM climate than under the PI climate and a doubled $CO_2$ climate. These studies suggest that a larger expansion of sea ice over the North Atlantic in the LGM would cause an increase in the recovery time of

the AMOC. Extensive sea ice covered the original deepwater formation regions and suppressed atmosphere–ocean heat exchange in the deepwater formation region (Oka et al. 2012, Sherriff-Tadano and Abe-Ouchi 2020), which made it difficult for the AMOC to recover after freshwater hosing had ceased (Bitz et al. 2007, Weber and Drijhout 2007). In contrast, Gong et al. (2013) compared the recovery time of the AMOC under PI, mid-glacial and LGM conditions in a comprehensive climate model and found that the recovery time was shortest in the mid-glacial case and longest in the PI case. They suggested that

greater subsurface ocean warming over the deepwater formation region, which affects ocean stratification (Mignot et al. 2007), was important in causing a shorter recovery time of the AMOC in the mid-glacial period. Furthermore, Goes et al. (2019) recently showed that the recovery time of the AMOC became shorter when they forced their EMIC with LGM winds compared with modern winds. Similarly, Sherriff-Tadano and Abe-Ouchi (2020) showed from sensitivity experiments with a comprehensive climate model that stronger surface winds shortened the duration of the weak AMOC state by increasing sea





surface salinity over the deepwater formation region. These results support the inference that changes in the background
      climate (e.g. ice sheet configurations and insolation) can modify the duration of stadials, although the processes and results
      may depend on the models used. However, in most studies, because the boundary conditions such as ice sheet configurations,
      $CO_2$ concentration and insolation are all modified at the same time, the impacts of individual boundary conditions on the
      durations of stadials and the recovery time of the AMOC remain elusive. A better understanding of the individual roles of
boundary conditions and their mechanism in modifying the recovery time is necessary to understand the changes in the
      durations of stadials across glacial periods, as well as to interpret model discrepancies.

      Previously, it has been shown that large Northern Hemisphere glacial ice sheets increase sea surface salinity over the North
      Atlantic Deepwater (NADW) formation region by increasing surface winds and decreasing precipitation (Eisenman et al. 2009,
Smith and Gregory 2012, Brady et al. 2013, Zhang et al. 2014a, Gong et al. 2015, Klockmann et al. 2016, Galbraith and de
      Lavergne 2019, Guo et al. 2019). In addition, it has been shown that stronger surface cooling by ice sheets increases the amount
      of sea ice in the NADW formation region and the Southern Ocean, the latter of which is induced by colder NADW outcropping
      in the Southern Ocean (Sherriff-Tadano et al. 2021). These results imply that differences in glacial ice sheets may play a role
      in modifying the durations of stadials during glacial periods. Recently, Sherriff-Tadano et al. (2021) performed simulations of
MIS3 and MIS5a and explored the impact of ice sheet differences on the AMOC and climate. In their simulations, differences
      in the ice sheets exerted small impacts on the vigorous mode of the AMOC, because of a compensational balance between the
      increase in sea surface salinity in the northern North Atlantic (strengthening effect) and the increase in sea ice in the North
      Atlantic and Southern Ocean (weakening effect). However, the impact of mid-glacial ice sheets on the duration of stadials and
      the recovery time of the AMOC remains elusive. Because the important processes affecting the stability of the AMOC may
differ between vigorous and weak AMOC modes (Buizert and Schmittner 2015, Lohmann 2019), a different response of the
      AMOC to ice sheet forcing under a weak AMOC state may be found.

      In this study, we explored the impacts of differences in the ice sheets between the MIS3 and MIS5a on the recovery time of
      the AMOC and the durations of stadials. For this purpose, we performed freshwater hosing experiments under three
background climates that have been simulated previously, MIS3, MIS5a and MIS3 with the ice sheet forcing of MIS5a
      (Sherriff-Tadano et al. 2021). By comparing the recovery time of the AMOC after the cessation of freshwater hosing in each
      experiment, we assessed the impact of the ice sheets on the recovery time of the AMOC. Furthermore, to explore the
      mechanism by which the ice sheets modify the recovery time of the AMOC, we performed partially coupled experiments. In
      these experiments, the atmospheric forcing, which is passed to the oceanic component of the model, was replaced with a
different forcing. By this method, individual effects of changes in surface wind, atmospheric freshwater flux, or surface cooling
      on the AMOC can be estimated (Mikolajewicz et al. 1997, Schmittner et al. 2002, Gregory et al. 2005, Sherriff-Tadano et al.
      2021).





This paper is organized as follows. Section 2 describes the model and the experimental design. In section 3, the impacts of the
ice sheet configurations on the recovery time of the AMOC and its mechanism are assessed. Section 4 discusses the results,
and section 5 presents the conclusion.

## 2. Methodology

### 2.1 Model

Numerical experiments were performed with the Model for Interdisciplinary Research on Climate 4m (MIROC4m; Hasumi
and Emori 2004), an atmosphere–ocean coupled general circulation model (AOGCM). This model consists of an atmospheric
general circulation model (AGCM) and an oceanic general circulation model (OGCM). The AGCM and OGCM include a land
surface model and a sea ice model, respectively. The AGCM solves the primitive equations on a sphere using a spectral method.
The horizontal resolution of the atmospheric model is ~2.8˚, and there are 20 layers in the vertical direction. The OGCM solves
the primitive equations on a sphere, with the Boussinesq and hydrostatic approximations adopted. The horizontal resolution is
~1.4˚ in longitude and 0.56˚–1.4˚ in latitude (latitudinal resolution is finer near the equator). There are 43 layers in the vertical
direction. Note that the coefficient of horizontal diffusion of the isopycnal layer thickness in the OGCM was slightly increased
to $700\,\mathrm{m}^2\,\mathrm{s}^{-1}$ compared with the original model version ($300\,\mathrm{m}^2\,\mathrm{s}^{-1}$) that was submitted to Paleoclimate Model Intercomparison
Project 2. These two model versions were referred to as Model B and Model A, respectively, by Sherriff-Tadano and Abe-
Ouchi (2020). Here, we used Model B. The model version used in this study has been used extensively for modern climate,
palaeoclimate (Obase and Abe-Ouchi 2019, O'ishi et al. 2021, Chan and Abe-Ouchi 2020) and future climate studies
(Yamamoto et al. 2015). It also reproduces the AMOC of the present, LGM (Sherriff-Tadano and Abe-Ouchi 2020), MIS3
and MIS5a (Sherriff-Tadano et al. 2021) reasonably well. See Hasumi and Emori (2004) and Chan et al. (2011) for detailed
information on the parameterizations used in the model.

### 2.2 Model simulations

This study was based on three climate simulations that have been performed previously (Sherriff-Tadano et al. 2021, Table 1).
The first climate simulation was that of MIS5a, which was forced with a $CO_2$ concentration of 240 ppm, insolation of 80 ka
and the ice sheet boundary configuration of 80 ka taken from Ice sheet model for Integrated Earth system Studies (IcIES, Abe-
Ouchi et al. 2007, Abe-Ouchi et al. 2013, Fig. 2a). The second and third climate simulations were those of MIS3, both of which
were forced with $CO_2$ of 200 ppm and insolation of 35 ka, but forced with ice sheets of either 36 ka (MIS3, Fig. 2b) or 80 ka
(MIS3-5aice). The volumes of the ice sheet were 40 metre sea level equivalent for 80 ka and 96 metre sea level equivalent for
36 ka (Abe-Ouchi et al. 2013). The Antarctic ice sheet was fixed to the modern configuration, and the Bering Strait remained
open in all experiments. For methane and other greenhouse gases, the concentration of the LGM was used (Dallenbach et al.
2000). These three simulations were initiated from a previous LGM experiment by Kawamura et al. (2017), and were integrated



for 2,000 years (MIS5a) or 3,000 years (MIS3 and MIS3-5aice). The decreasing trends of deep ocean temperature of the last
100 years were 0.002 ˚C in MIS5, 0.011 ˚C in MIS3 and 0.007 ˚C in MIS3-5aice, respectively (Sherriff-Tadano et al. 2021).

To cause drastic weakening of the AMOC and shift the climate into stadial, a freshwater flux of 0.1 Sv was applied
uniformly over the northern North Atlantic (50˚–70˚ N) for 500 years (Fig. 3). Subsequently, the freshwater flux was stopped
and the experiments were further integrated for 1,000 years to assess the dependence of the recovery time on the background
climate. These experiments are named MIS3H, MIS5aH and MIS3-5aiceH, respectively. The impact of the mid-glacial ice
sheet on the duration of stadials was assessed by comparing the recovery time of the AMOC between MIS3H and MIS3-
5aiceH. The effect of the differences in $CO_2$ and insolation could be assessed by comparing the recovery time of the AMOC
between MIS3-5aiceH and MIS5aH.

### 2.3 Partially coupled experiments

To clarify the mechanisms by which glacial ice sheets modify the recovery time of the AMOC, partially coupled experiments
were conducted (Table 2). In these experiments, the atmospheric forcing – wind stress and atmospheric freshwater flux
(precipitation, evaporation and river runoff) – that drove the ocean was replaced with monthly climatology. The heat flux was
unchanged in these experiments, as it was strongly coupled with the sea surface temperature and fixing the surface heat
conditions has an unrealistic impact on the AMOC (Schmittner et al. 2002, Gregory et al. 2005, Marozke 2012). Atmospheric
forcing was replaced with monthly climatology of the last 100 years of the hosing period in each experiment. Thus, this forcing
did not include atmospheric noise, which itself can induce a mode shift of the AMOC (Ganopolski and Rahmstrof 2001,
Kleppin et al. 2015). Understanding the role of atmospheric noise is beyond the scope of this study, but should be explored in
other studies.

Five partially coupled experiments were conducted under MIS3H and MIS3-5aiceH (Table 2). All of the experiments were
initiated from the first year of the cessation of freshwater hosing, which corresponded to the period when the climate and
AMOC had settled to the stadial state (see Figs. 11 and 12). The first two experiments served as a validation of the method;
the atmospheric forcing was replaced with the climatology in MIS3H and MIS3-5aiceH. These experiments are named PC-
MIS3H and PC-MIS3-5aiceH, respectively. We regarded the method as valid when these experiments reproduced the general
difference of MIS3H and MIS3-5aiceH. In the other three experiments, the atmospheric forcing was replaced with different
forcing. In PC-MIS3H_wind, the surface wind stress of MIS3-5aiceH was applied to MIS3H. In PC-MIS3H_water, the
atmospheric freshwater flux of MIS3-5aiceH was applied to MIS3H. In PC-MIS3H_windwater, the atmospheric freshwater
flux and surface wind stress of MIS3-5aiceH were applied to MIS3H. From these experiments, the impact of differences in the
wind was estimated as the difference between PC-MIS3H_wind and PC-MIS3H, the impact of differences in the atmospheric
freshwater flux was estimated as the difference between PC-MIS3H_water and PC-MIS3H, and the impact of differences in
the surface cooling was estimated as the difference between PC-MIS3H_windwater and PC-MIS3-5aiceH. Note that the effect



of surface cooling (heat flux) was estimated as a residual, following previous studies (Gregory et al. 2005). The surface cooling effect included the effects of changes in freshwater flux of sea ice.

## 3. Results

Simulated climates of unperturbed MIS5a, MIS3 and MIS3-5aice are displayed in Figs. 3–5. The simulated global air temperatures were 10.6 ˚C in MIS5a, 7.9 ˚C in MIS3 and 8.9 ˚C in MIS3-5aice (Sherriff-Tadano et al. 2021). The maximum strength of the AMOC was 18.4 Sv in MIS5a, 15.6 Sv in MIS3 and 15.1 Sv in MIS3-5aice (Fig. 4). The slightly weaker AMOC in MIS3 than in MIS5a is consistent with a reconstruction based on $^{231}$Pa/$^{230}$Th (Bohm et al. 2015). Associated with the vigorous AMOC, deepwater formed in the Greenland Sea and the Irminger Sea, and most parts of the northern North Atlantic remained ice-free in all experiments (Fig. 5). These characteristics are consistent with proxy data suggesting ice-free conditions in the Norwegian Sea during interstadials (Dokken et al. 2013, Sadazki et al. 2019).

### 3.1 Responses to freshwater hosing

To shift the climate and AMOC into stadial states, freshwater hosing experiments were performed under these background climate conditions. These experiments all showed drastic weakening of the AMOC in response to hosing (Figs. 3 and 4). The strength of the AMOC decreased to 3 Sv in MIS5aH and MIS3-5aiceH, and decreased to 5 Sv in MIS3H. In addition, the Antarctic bottom water further penetrated into the North Atlantic compared with unperturbed conditions (Fig. 4). Associated with the weakening of the AMOC, sea ice expanded farther south and reached 50˚ N (Fig. 5). As a result, the deepwater formation region was covered by sea ice and the sea surface temperature over the northern North Atlantic was drastically reduced (Fig. 6). In addition, the surface salinity decreased drastically at high latitudes (Fig. 6) because of freshwater hosing, cessation of convective mixing and a reduction in northward salt transport by the AMOC. In contrast, the subsurface ocean temperature increased at high latitudes because of the suppression of convective mixing. These characteristics are consistent with proxies (Rasmussen and Thomsen 2004, Dokken et al. 2013). In the tropics, the subsurface ocean temperature and salinity increased because of the weakening of the northward transport of heat and salt by the AMOC (Gong et al. 2013).

The weakening of the AMOC and the expansion of sea ice induced drastic cooling over Greenland (Fig. 7a). In particular, the February temperature decreases by 12 ˚C in MIS3H, 10 ˚C in MIS3-5aiceH and 12 ˚C in MIS5aH, which are within the range of ice core data (Kindler et al. 2014). Over the Antarctic, the temperature increases by 1–2 ˚C because of the bipolar seesaw (Kawamura et al. 2017). In terms of precipitation (Fig. 7b), the model reproduced a southward shift of the tropical rain belt (Wang et al. 2004) and weakening of the Indian monsoon (Deplazes et al. 2014). Therefore, the model reproduced the overall characteristics of the climate shift into stadial reasonably well.

### 3.2 Recovery

The AMOC recovered from the weak state to the vigorous state in all experiments after the cessation of freshwater hosing



(Fig. 3), although the recovery time differed among the experiments. In MIS5aH, the AMOC started to recover abruptly 200 years after the cessation of hosing. In MIS3H, the AMOC first increased to 4.5 Sv over the first 80 years and then intensified

abruptly to the vigorous mode of 7.1 Sv in 70 years (the recovery speed nearly doubled compared with the first 80 years). The recovery time was slightly shorter in MIS3H compared with MIS5aH, which is consistent with the ice core data showing slightly shorter durations of stadials during MIS3 compared with MIS5a-d (Buizert and Schmittner 2015). In contrast, the recovery time was much longer in MIS3-5aiceH; it took approximately 600 years to start the drastic recovery. Before that, the AMOC recovered gradually by 3 Sv over the first 560 years. Around model year 1065, the AMOC was abruptly enhanced and

its strength reached 10 Sv. The strength of the AMOC once decreased to 7 Sv, although 100 years after the first abrupt strengthening, the AMOC started to recover abruptly to the interstadial state. These results reveal three important points. First, the larger mid-glacial ice sheets in MIS3 compared with those of MIS5a shortened the recovery time of the AMOC. Second, the lowering of the $CO_2$ and the changes in insolation from MIS5aH to MIS3-5aiecH contributed to the increase in the recovery time of the AMOC in our experiments. This is consistent with other studies showing an increase in the durations of stadials

under lower $CO_2$ concentrations (Brown and Galbraith 2016, Klockmann et al. 2018). Third, the recovery time of the AMOC could not be predicted based on the original strength of the AMOC because the recovery time was shorter in MIS3H compared with MIS5aH, even though the original AMOC was weaker. In MIS3H, the effect of the glacial ice sheet was stronger than that of $CO_2$, and thus caused shortening of the recovery time compared with MIS5aH. Below, we further compare the recovery process in MIS3-5aiceH and MIS3H to understand how glacial ice sheets modify the recovery time, which remained unclear

in previous studies.

To understand the recovery process of MIS3-5aiceH, time series of sea ice, deepwater formation, surface salinity and subsurface ocean temperature were analyzed (Renold et al. 2010, Vettoretti and Peltier 2016, Brown and Galbraith 2016). Figure 8 shows time series of these variables in the Irminger Sea (35–25˚ W, 55–63˚ N) and Greenland Sea (1˚ W–5˚ E,

65˚N–70˚N), where deepwater formed at the onset of the abrupt recovery of the AMOC. In MIS3–5aiceH, after the cessation of hosing, surface salinity and density first increased drastically in the Irminger Sea and Greenland Sea (red line in Fig. 8), followed by a gradual increase afterwards. In association, the AMOC strengthened slightly and increased the northward transport of salt and heat, which induced a slight increase in the subsurface temperature and surface salinity and a decrease in sea ice. Formation of deepwater occurred in the Irminger sea approximately 300 years after the cessation of hosing, but the

AMOC did not recover at this point because the surface salinity and subsurface ocean temperature were not sufficiently high to maintain convection. Four hundred years after the cessation of hosing, the surface salinity and sea ice thickness reached a quasi-equilibrium state, whereas the subsurface temperature continuously increased. When the subsurface ocean warmed sufficiently, vigorous convective mixing initiated again in the Irminger Sea (Figs. 8 and 9, regions circled by black contours). As a result, a positive salinity anomaly spread over the subpolar gyre regions (Fig. 9), which caused a second

deepwater formation in the north-western North Atlantic in the Greenland Sea, where the surface salinity was sufficiently high and subsurface ocean sufficiently warm (Figs. 8 and 9). These deepwater formations did not occur continuously and





they ceased once, possibly associated with decadal variability in deepwater formation (Oka et al. 2006). Note that the emergence of enhanced decadal variability prior to the full AMOC recovery is in line with the observation of early warning signals for DO events (i.e., signs of a tipping point) in a high-resolution ice core record (Boers 2018). However, the

deepwater formation in the Greenland Sea induced southward flow through the Denmark Strait in the deep ocean and enhanced the AMOC via downward flow along the slope (Reynolds et al. 2010). As a result, a compensational northward surface flow transported salt into the deepwater formation and caused a second occurrence of convection in the Greenland Sea (Fig. 9, years 1075 to 1079). Subsequently, the AMOC recovered abruptly to its original strength with an overshoot (Fig. 3). These recovery processes show that the balance of sea ice thickness, sea surface salinity and subsurface ocean

temperature determined the recovery time of the AMOC in this experiment.

In contrast, the recovery process differed in MIS3H (black line in Fig. 8). At first, during the hosing period, sea surface salinity was higher and sea ice thickness was thinner compared with MIS3-5aiceH, which were favourable conditions to induce deepwater formation. Then, after the cessation of freshwater hosing, deepwater formation initiated, triggered by the initial

increase of surface salinity. Because the surface salinity was sufficiently high in the weak phase of the AMOC, deepwater could form continuously. As a result, vertical mixing occurred continuously and further increased surface salinity and decreased sea ice thickness over the Irminger Sea and Greenland Sea, causing gradual strengthening of the AMOC. Then, the gradual increase in the AMOC induced a further increase in surface salinity and a decrease in sea ice (Fig. 8) over the Greenland Sea. As a result, 80 years after the cessation of hosing, deepwater formation initiated in the Greenland Sea, and the AMOC

abruptly recovered. Thus, in MIS3H, changes in the surface salinity and sea ice thickness played a larger role in controlling the recovery time of the AMOC, whereas the changes in subsurface ocean temperature played a minor role in the recovery.

The above analysis suggests that the differences in surface salinity and sea ice between MIS3H and MIS3-5aiceH under the hosing phase caused the difference in the recovery time; in MIS3H, surface salinity was higher and sea ice thickness was

thinner compared with MIS3-5aiceH, which favoured a shorter recovery time. The differences in sea ice and surface salinity may be attributed to a difference in the surface wind (Sherriff-Tadano et al. 2018). Figure 10a and d show how the surface wind differed in the two experiments. Anomaly fields in Fig. 10d reveal the enhancement of cyclonic wind over the northern North Atlantic and southward displacement of the westerly winds in MIS3H compared with MIS3-5aiceH, which were induced by the topography of the Laurentide ice sheet (Pausata et al. 2011, Sherriff-Tadano et al. 2021). With the southward-shifted

westerly wind and strong northerly wind over the western North Atlantic, less sea ice was transported to the deepwater formation region in MIS3H (Fig. 10c, f). Therefore, even though the atmosphere was colder, less sea ice existed over the deepwater formation region. In terms of surface salinity, the wind intensified the Ekman upwelling and gyre circulation that transport warm and saline water to the deepwater formation region and support convection through increasing the surface salinity and decreasing the sea ice (Fig. 10b, e, Montoya et al. 2011, Muglia and Schmittner 2015, Sherriff-Tadano et al. 2018).

In fact, a positive wind stress curl was larger in the subpolar region and the Irminger Sea in MIS3H compared with MIS3-



5aiceH (Fig. 10d). Therefore, differences in winds over the northern North Atlantic seemed to contribute to the difference in the recovery time between the two experiments by modulating the surface salinity and sea ice in the stadial period.

### 3.3 Partially coupled experiments

To clarify the impact of differences in surface wind between MIS3H and MIS3-5aiceH on the recovery time of the AMOC,

partially coupled experiments were conducted from the first year after the cessation of freshwater hosing (Fig. 11). First, the reproducibility of the original experiments by the partially coupled experiments was assessed. In PC-MIS3H and PC-MIS3-5aiceH, the recovery time was slightly shorter compared with the corresponding original experiments. In particular, the recovery time was 200 to 300 years shorter in PC-MIS3-5aiceH compared with MIS3-5aiceH. This was related to the removal of sub-monthly variations in wind stress (Sherriff-Tadano et al. 2021); removal of these variations caused thinning of sea ice

in the centre of the subpolar region by reducing sea ice transport in this region (Fig. 12b, c) and created favourable conditions for deepwater to form. Nevertheless, even though PC-MIS3-5aiceH underestimated the recovery time, PC-MIS3H and PC-MIS3-5aiceH at least reproduced the main difference of the recovery time between MIS3H and MIS3-5aiceH.

Next, the effect of surface wind on the recovery time of the AMOC was explored. When the surface winds of the MIS5a ice

sheet (MIS3-5aiceH) were applied to PC-MIS3H (PC-MIS3H_wind), the AMOC did not start to recover in the first 100 years, as seen in MIS3H. This was related to the weaker surface wind, which reduced the wind-driven oceanic transport of salt into the deepwater formation and caused a decrease of sea surface salinity there. Thus, partially coupled experiments showed that the stronger wind in MIS3H created favourable conditions to cause an earlier recovery of the AMOC. This was also confirmed by another sensitivity experiment showing earlier recovery of the AMOC when the surface wind of MIS3H was applied to PC-

MIS3-5aiceH (not shown).

Interestingly, the AMOC did not recover in PC-MIS3H_wind during the integration. A similar feature was also observed in PC-MIS3H_windwater, where the model was forced with the heat flux of the MIS3 ice sheet (MIS3H) and the surface wind and atmospheric freshwater flux of the MIS5a ice sheet (MIS3-5aiceH). This long stadial state was caused by the very thick

sea ice over the deepwater formation region, associated with stronger surface cooling by the MIS3 ice sheet (Fig. 12b, d). After the cessation of freshwater hosing and the replacement of the surface wind, the sea surface salinity as well as the subsurface ocean temperature increased gradually in PC-MIS3H_wind and PC-MIS3H_windwater. However, the thick sea ice over the deepwater region prevented the initiation of deepwater formation and maintained the weak AMOC (Loving and Vallis 2005, Bitz et al. 2007, Oka et al. 2012, Sherriff-Tadano et al. 2021). This result shows that the cooling effect of the MIS3 ice sheet

played a role in increasing the recovery time of the AMOC by increasing sea ice over the deepwater formation region.

Lastly, the effects of differences in atmospheric freshwater flux on the recovery time of the AMOC were explored for completeness. When the atmospheric freshwater flux of the MIS5a ice sheet (MIS3-5aiceH) was applied to PC-MIS3H (PC-




MIS3H_water), the recovery time of the AMOC increased slightly. This was associated with a decrease of sea surface salinity

over the deepwater formation region (Fig. 12a), which was linked to the northward shift of the rain belt in the mid-latitudes
caused by the smaller ice sheet (Eisenman et al. 2009). Therefore, the larger (smaller) MIS3 (MIS5a) ice sheet reduced
(increased) the recovery time of the AMOC by reducing (increasing) the input of atmospheric freshwater flux over the
deepwater formation region. Nevertheless, the differences in atmospheric freshwater flux had less impact on the duration of
the recovery compared with the effect of wind in these experiments. To summarize, the shorter recovery time in MIS3H

compared with MIS3-5aiceH was a result of the dominance of the surface wind effect caused by larger ice sheets, which
promoted the recovery of the AMOC, compared with the surface cooling effect, which promoted increase in the recovery time
of the AMOC.

### 4. Discussion

Our results show that the recovery time of the AMOC largely depend on the background climate. In MIS3H, the AMOC started

to recover soon after the cessation of freshwater hosing, whereas in MIS3-5aiceH, the AMOC first recovered gradually for
several hundred years and then recovered abruptly. It was found that the difference in surface wind played a role in causing
the difference between MIS3H and MIS3-5aiceH. The cyclonic surface wind at mid-high latitudes was stronger in MIS3H
than in MIS3-5aiceH. In addition, a strong northerly wind anomaly was induced over the western North Atlantic. As a result,
the wind-driven transport of salt to the deepwater formation region was larger and wind-driven sea ice transport smaller in

MIS3H compared with MIS3-5aiceH. This led to higher surface salinity and thinner sea ice thickness over the deepwater
formation region, which increased the probability of the recovery of the AMOC. Thus, the changes in the surface wind caused
by the glacial ice sheet could contribute to a shorter stadial during MIS3 compared with MIS5.

Previous studies have shown that the subsurface ocean temperature (Mignot et al. 2007, Gong et al. 2013), freshwater transport

by the AMOC (de Vreis and Weber 2005, Weber and Drijfhout 2007, Liu et al. 2014) and surface winds (Goes et al. 2019)
affect the recovery time of the AMOC. Our analysis of these parameters in hosing experiments showed results consistent with
these studies. With respect to subsurface ocean temperature, the subsurface ocean temperature anomaly was larger in MIS3H
than in MIS3H-5aiceH, which favoured early recovery of the AMOC by destabilizing the water column in the deepwater
formation region (Gong et al. 2013). With respect to freshwater transport by the AMOC, our analysis showed a larger amount

of freshwater transport into the Atlantic in MIS3H than in MIS3-5aiceH (0.073 Sv and 0.017 Sv, respectively, before
freshwater hosing). Thus, the results were also consistent with previous studies in that the experiment in which the AMOC
transported more freshwater in the Atlantic recovered more quickly. Nevertheless, as shown in the partially coupled
experiments, the AMOC could not recover in PC-MIS3H_wind when the surface wind was weak over the deepwater formation
region. With respect to wind forcing, Goes et al. (2019) showed that the stronger surface wind in the LGM caused a shorter

recovery time of the AMOC compared with that from the modern climate. Our study is also in line with their study in that the
stronger surface wind in MIS3 compared with MIS5a induced by ice sheet differences caused a shorter recovery time of the





AMOC. Therefore, together with Goes et al. (2019), this study reveals another important control on the recovery time of the AMOC: differences in localities of winds in the deepwater formation region. In this regard, this study supports the conclusion of Weber and Drijfhout (2007) and Bitz et al. (2007) that differences in atmospheric conditions play a role in controlling the

recovery time of the AMOC.

Our findings can be used to interpret model discrepancies. Gong et al. (2013) showed that the recovery time of the AMOC was shorter under mid-glacial and LGM conditions compared with the PI climate, whereas Weber and Drijhout (2007) and Bitz et al. (2007) show that the recovery time was longer under LGM conditions compared with PI conditions. In these studies, all of

the boundary conditions (e.g. glacial ice sheets and $CO_2$) were modified; thus, the reason for differences between the models remains elusive. Based on this study, we suggest that the wind effect of the glacial ice sheets played the dominant role in the study of Gong et al. (2013), whereas the sea ice effect caused by lowering of the $CO_2$ concentration and by the glacial ice sheet played a larger role in the studies of Weber and Drijhout (2007) and Bitz et al. (2007). In fact, the surface winds were strongest in the mid-glacial experiment compared with the other experiments of Gong et al. (2013, 2015). In contrast, the surface winds

were not stronger in the LGM simulations compared with the PI simulation by Bitz et al. (2007, Otto-Bliesner et al. 2006), even under the existence of glacial ice sheets. Although the cause of the difference in surface wind remains elusive, differences in the strength of the surface winds between models may have caused the difference in the recovery time. Because Weber and Drihout (2007) used an EMIC, the model may have underestimated the wind change caused by the glacial ice sheets. Therefore, the wind effect may not have had a strong impact, and thus the sea ice effect played the dominant role.


Ice core studies have recently suggested a possibility that the relation between the background climate and the durations of climate states can differ between interstadials and stadials; although the durations of both interstadials and stadials are generally affected by global temperatures and surface cooling (Buizert and Schmittner 2015, Kawamura et al. 2017, Lohmann and Ditlevsen 2019), the durations of stadials may be affected by additional conditions over the Northern Hemisphere (Lohmann

2019) when the global climate is generally cold. A similar feature was also observed in climate model simulations of Sherriff-Tadano et al. (2021) and this study. For example, Sherriff-Tadano et al. (2021) showed that differences in the vigorous AMOC between MIS5a and MIS3 were mainly caused by the differences in $CO_2$. In their simulations, ice sheet differences had small impacts on the vigorous AMOC because of compensational balance between the strengthening effect of surface wind and the weakening effect of sea ice increase in the Northern and Southern Hemispheres. In contrast, in the hosing experiments of the

present study, the effect of surface wind by the larger MIS3 ice sheets appeared to be stronger compared with stronger surface cooling by the ice sheets and lower $CO_2$, causing shortening of the stadials in MIS3 compared with MIS5a. These results support the findings of ice core studies and suggest that the relation between the background climate and the durations of climate states can differ between interstadials and stadials.

Although the expansion of glacial ice sheets from MIS5a to MIS3 could have contributed to short stadials during the mid-glacial period, there is another unsolved problem: why were stadials very long during MIS2 and MIS4, when the glacial ice sheets were at their largest size (McManus et al. 1999, Buizert and Schmittner 2015, Kawamura et al. 2017)? During these periods, summer insolation over the North Atlantic was very low; therefore, this may be important. In fact, Turney et al. (2015) showed that lowering of the obliquity in MIS2 weakened the AMOC by increasing sea ice in the North Atlantic. In addition,

very strong surface cooling by the glacial ice sheets may have caused long stadials. In fact, we found that the strengthening of surface cooling by the larger ice sheets could increase the recovery time of the AMOC by increasing the amount of sea ice over the deepwater formation region. If there was a shift from a wind-dominated ice sheet effect, which shortened the recovery time of the AMOC, to a surface cooling-dominated ice sheet effect, the large ice sheets during MIS2 and MIS4 could contribute to the very long stadials. Further investigations of the roles of insolation and the ice sheet effect will be important for better

understanding the glacial AMOC as well as interpreting the controlling parameters changing the duration of stadials over the glacial period.

Lastly, drastic weakening of the AMOC was induced by freshwater hosing in this study. However, recent studies have shown that the large-scale freshwater hosing was a result of weakening of the AMOC, rather than the cause of the drastic weakening of the AMOC (Alvarez-Solas et al. 2011, Barker et al. 2015). Nevertheless, the main point of our results is that once the AMOC

was weakened by external forcing, the recovery time of the AMOC differed because of the ice sheet configurations. Thus, the external forcing that induced the weakening of the AMOC did not have to be a large discharge from the ice sheet and could have been other forcing, such as a small amount of freshwater flux from the ice sheet, or perhaps volcanic eruptions. Thus, our results are applicable for DO cycles forced by external forcing. However, previous studies have shown that DO cycles may be excited by internal oscillation of the atmosphere–sea ice–ocean system (Arzel et al. 2010, Peltier and Vettoretti 2014, Vettoretti

and Peltier 2016, Brown and Galbraith 2016, Klockmann et al. 2018, Sherriff-Tadano and Abe-Ouchi 2020). Although we have not explicitly investigated this case, we speculate that stronger winds in the northern North Atlantic could increase the probability of deepwater formation during stadials by modifying the balance of sea ice, surface salinity and subsurface ocean temperature. Nevertheless, it is important to assess our findings in this case as well.

**5. Conclusion**

To understand the reason why the durations of stadials were shorter during MIS3 compared with MIS5 despite the generally colder climate in MIS3, we explored the impact of the mid-glacial ice sheets on the durations of stadials. For this purpose, we conducted freshwater hosing experiments with the MIROC4m AOGCM under MIS3 and MIS5a conditions. Furthermore, to extract the impact of the difference in the glacial ice sheets on the recovery time of the AMOC, a sensitivity experiment was performed, which was forced with the MIS5a ice sheet under MIS3 $CO_2$ and insolation conditions (MIS3-5aice). The ice sheets

of MIS3 and MIS5a were taken from an ice sheet model, which reproduced the evolution of the ice sheets over the last 400,000 years (Abe-Ouchi et al. 2013). Freshwater hosing of 0.1 Sv over the northern North Atlantic induced collapse of the AMOC

and southward expansion of sea ice, which covered the deepwater formation region in all experiments. After the cessation of freshwater hosing, the AMOC recovered in all experiments, which was associated with the initiation of deepwater formation in both the Irminger Sea and the Greenland Sea. However, the recovery time of the AMOC differed among the experiments;

following the cessation of freshwater hosing, recovery started after 80 years in MIS3, after approximately 200 years in MIS5a, and after approximately 600 years in MIS3-5aice. The slightly shorter recovery time in MIS3 compared with MIS5a was consistent with the ice core data. The sensitivity experiment (MIS3-5aice) extracting the effect of the mid-glacial ice sheet showed that a larger glacial ice sheet caused a shorter recovery time in MIS3, whereas lowering of the $CO_2$ concentration and changes in insolation caused an increase of the recovery time. The partially coupled experiments further showed that stronger

surface winds over the North Atlantic shortened the recovery time by increasing the surface salinity and decreasing the sea ice amount in the deepwater formation region. In contrast, we also found that the surface cooling caused by larger ice sheets tended to increase the recovery time of the AMOC by increasing the sea ice thickness over the North Atlantic. In our simulation, the effect of surface winds appeared to be stronger than the effect of surface cooling, thus causing a shortening of the recovery time of the AMOC. Therefore, our results suggest that the expansion of glacial ice sheets played a role in reducing the duration

of stadials during MIS3 and thus could contribute to the frequent DO cycles during MIS3 when the effect of surface winds dominated. Nevertheless, the effect of surface cooling may be important when the long stadials during the MIS2 and MIS4 and the model discrepancies are considered.

**Code and data availability**

The MIROC code associated with this study is available to those who conduct collaborative research with the model users

under license from copyright holders. The code of partially coupled experiments is available from the corresponding author (S. S.-T.) upon reasonable request. The simulation data will be available from https://ccsr.aori.u-tokyo.ac.jp/~tadano/.

**Author contribution**

S. S.-T. performed the climate model simulation and analyzed the results with the assistance of A. A.-O. S. S.-T. performed the partially coupled experiments with the assistance of A. O. The manuscript was written by S. S.-T. with contributions from

all authors.

**Competing interest**

The authors declare no competing interests.

**Acknowledgements**





We thank Masahide Kimoto, Hiroyasu Hasumi, Masahiro Watanabe, Ryuji Tada, Masakazu Yoshimori and Takashi Obase
for constructive discussion. The model simulations were performed on the Earth Simulator 3 at JAMSTEC. This study was
supported by the Program for Leading Graduate Schools, MEXT, Japan, and JSPS KAKENHI Grant Numbers 15J12515,
17H06104, 17H06323 and 20K14552. T.M. acknowledges funding by the Volkswagen Foundation. We thank Sara J. Mason
for editing a draft of this manuscript.

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






**Table 1: Forcing and boundary conditions of the climate simulations.**

| Name | CO2 | Ice sheet | Obliquity | Precession | Ecc |
|---|---|---|---|---|---|
| MIS5aH | 240 ppm | 80ka | 23.175 | 312.25 | 0.0288 |
| MIS3H | 200 ppm | 36ka | 22.754 | 251.28 | 0.0154 |
| MIS3-5aiceH | 200 ppm | 80ka | 22.754 | 251.28 | 0.0154 |

**Table 2: List of partially coupled experiments. In each of these experiments, atmospheric forcing was replaced by the climatology of the specified experiment.**

| Name | Surface wind | Atmos. Fw forcing | Surface cooling |
|---|---|---|---|
| PC-MIS3H | MIS3 | MIS3 | MIS3 |
| PC-MIS3-5aiceH | MIS3-5aice | MIS3-5aice | MIS3-5aice |
| PC-MIS3H_wind | MIS3-5aice | MIS3 | MIS3 |
| PC-MIS3H_water | MIS3 | MIS3-5aice | MIS3 |
| PC-MIS3H_windwater | MIS3-5aice | MIS3-5aice | MIS3 |


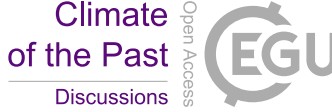

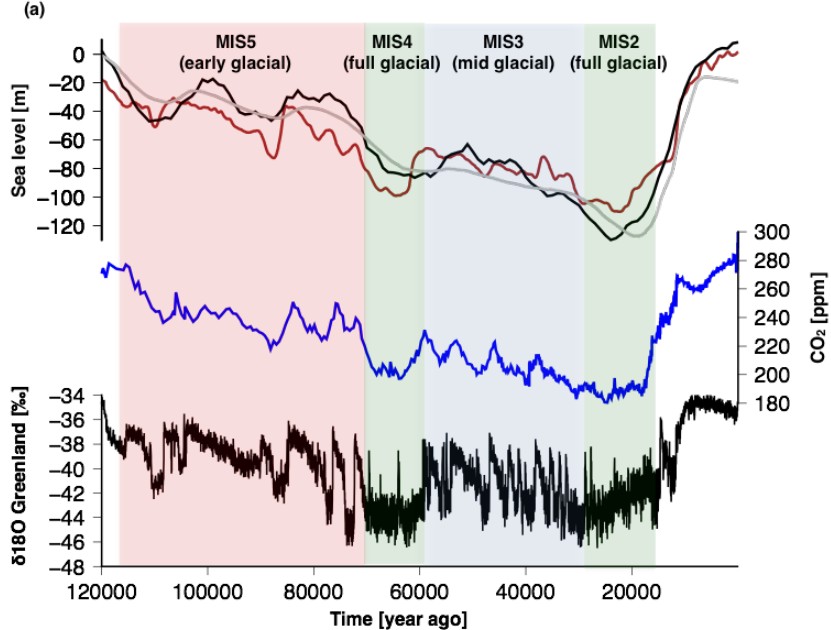

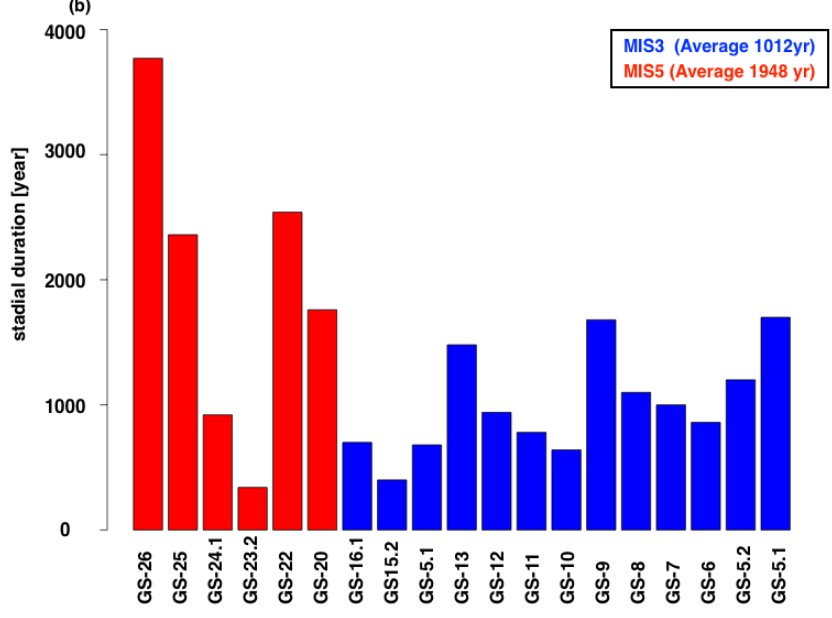







**Figure 1: (a) Changes in ice volume (sea level, black: Spratt and Lisiecki 2016, brown: Grant et al. 2012, grey: Abe-Ouchi et al. 2013), CO$_2$ (Beleiter et al. 2015) and Greenland ice core δ$^{18}$O (a proxy of temperature, Rasmussen et al. 2013) over the last glacial–interglacial cycle. (b) Duration of stadials during MIS5 and MIS3. Colour shading in (a) separates the period within one glacial cycle. Green: MIS2, MIS4; blue: MIS3; and red: MIS5a-d. During MIS2 and MIS4, the glacial ice sheets were most extensive and the CO$_2$ concentration was low. During MIS3, the glacial ice sheets were relatively small and the CO$_2$ concentration was higher than those of MIS2 and MIS4. During MIS5a-d, the glacial ice sheets were smallest and the CO$_2$ concentration was highest among the glacial periods.**

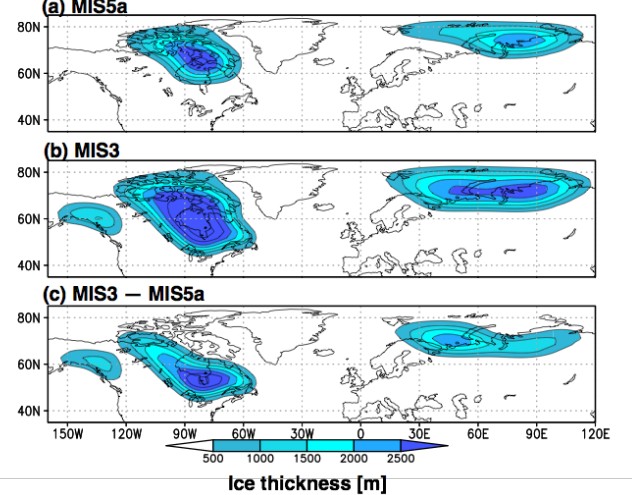

**Figure 2: Ice sheet thickness of (a) MIS5a (80 ka), (b) MIS3 (36 ka) and (c) MIS3 minus MIS5a. Results from an ice sheet model are presented (Abe-Ouchi et al. 2013). These ice sheet configurations were used for climate model simulations.**



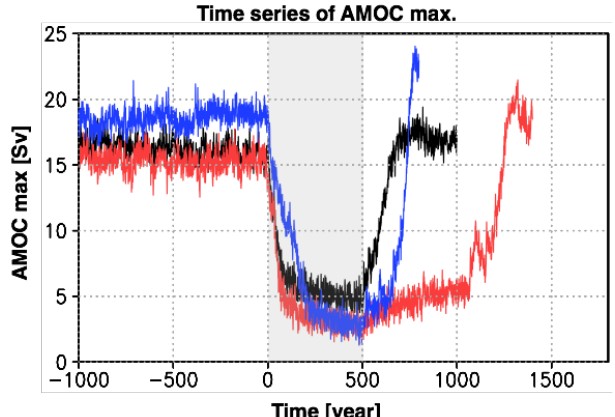

**Figure 3: Time series of the strength of the AMOC under freshwater hosing. Freshwater of 0.1 Sv was released to 50–70˚ N in the North Atlantic for 500 years from year 1 (grey shaded period). Black: MIS3H, blue: MIS5aH, and red: MIS3-5aiceH.**

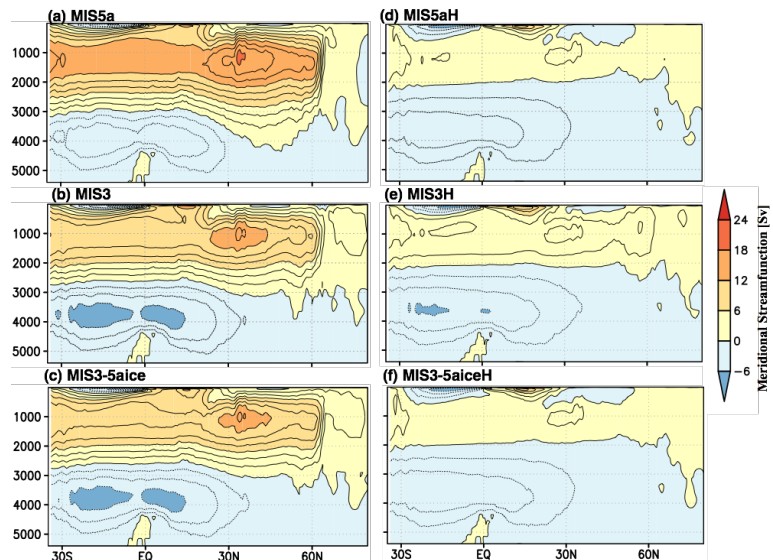

**Figure 4: Climatology of the meridional streamfunction [Sv = $10^6$ m$^3$ s$^{-1}$] in the Atlantic. (a) MIS5a, (b) MIS3, (c) MIS3-5aice, (d)**
**MIS5aH, (e) MIS3H, and (f) MIS3-5aiceH. The last 100 years are used for analysis.**





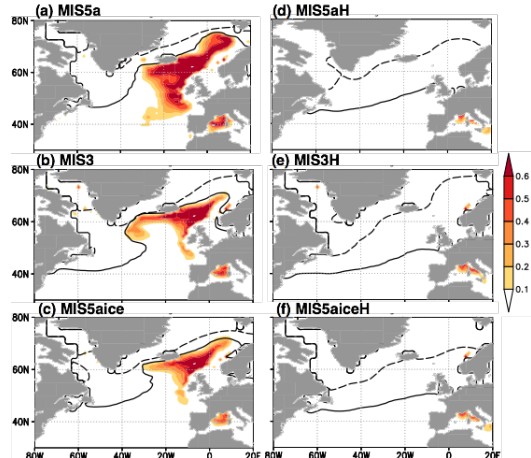

**Figure 5: Sea ice edge (contour, 50% concentration) for February (solid) and August (dashed) and deepwater formation region shown in annually averaged frequency of the convective adjustment at 300 m depth (colour). (a) MIS5a, (b) MIS3, (c) MIS3-5aice, (d) MIS5aH, (e) MIS3H, and (f) MIS3-5aiceH. The last 100 years are used for analysis.**

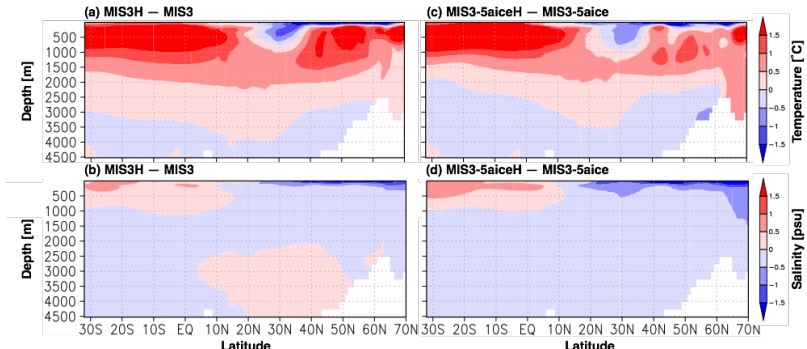


**Figure 6: Oceanic responses to freshwater hosing in MIS3 (a, b) and MIS3-5aice (c, d). Figures on the top show annual mean ocean temperature anomalies [°C, colour], and figures on the bottom show salinity anomaly [psu, colour]. Differences between the last 100 years of hosing and the last 100 years before hosing are shown.**





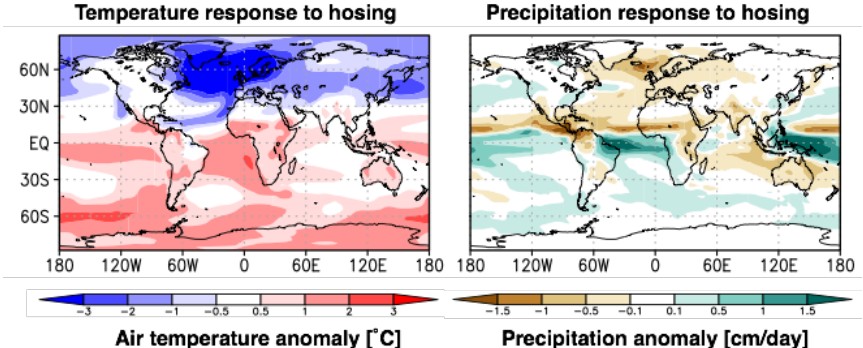

**Figure 7: Atmospheric responses to freshwater hosing in MIS3-5aiceH. (Left) Annual mean surface air temperature [°C, colour] and (right) precipitation anomaly [cm/day, colour]. Differences between the last 100 years of hosing and last 100 years before hosing are shown.**

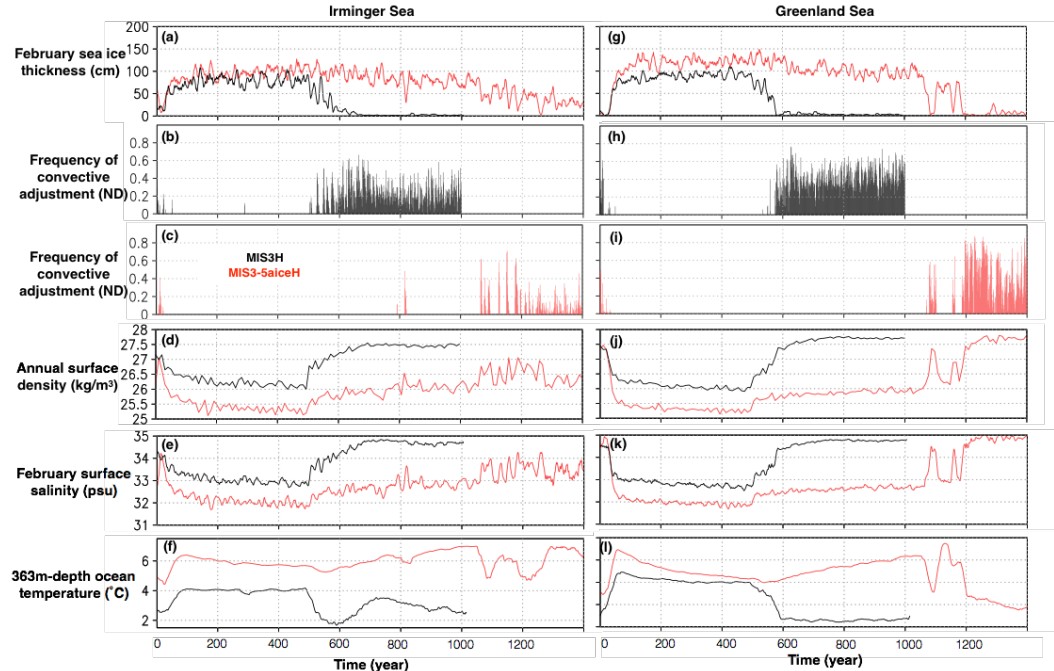

**Figure 8: Temporal evolution of oceanic variables in the Irminger Sea (35–25° W, 55–63° N, left) and Greenland Sea (1° W–5° E, 65–70° N, right) for MIS3H (black) and MIS3-5aiceH (red). (a, g) February sea ice thickness [cm]. (b, h) annual average frequency of convective adjustment of MIS3H. Panels (c, i) same as (b, h) but for MIS3-5aiceH. (d, j) Annual mean surface density [kg m$^{-3}$].**



(e, k) February surface salinity [psu] and (f, l) annual mean subsurface ocean temperature [˚C]. Except for (b), (c), (h) and (i), 11-year running means are shown. Freshwater flux of 0.1 Sv is applied during year 0 to 500.

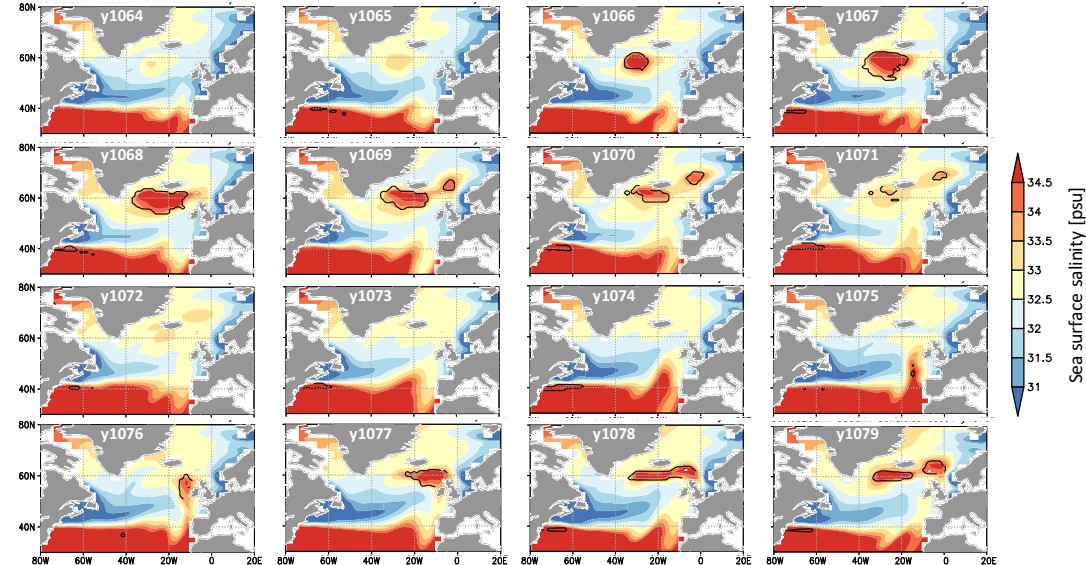


**Figure 9: Spatial maps of the temporal evolution of surface salinity [colour, psu] and the frequency of convective adjustment at 300-metre depth [black contours, non-dimensional) in MIS3-5aiceH. Annual means are shown. The value of the contour is 0.1.**

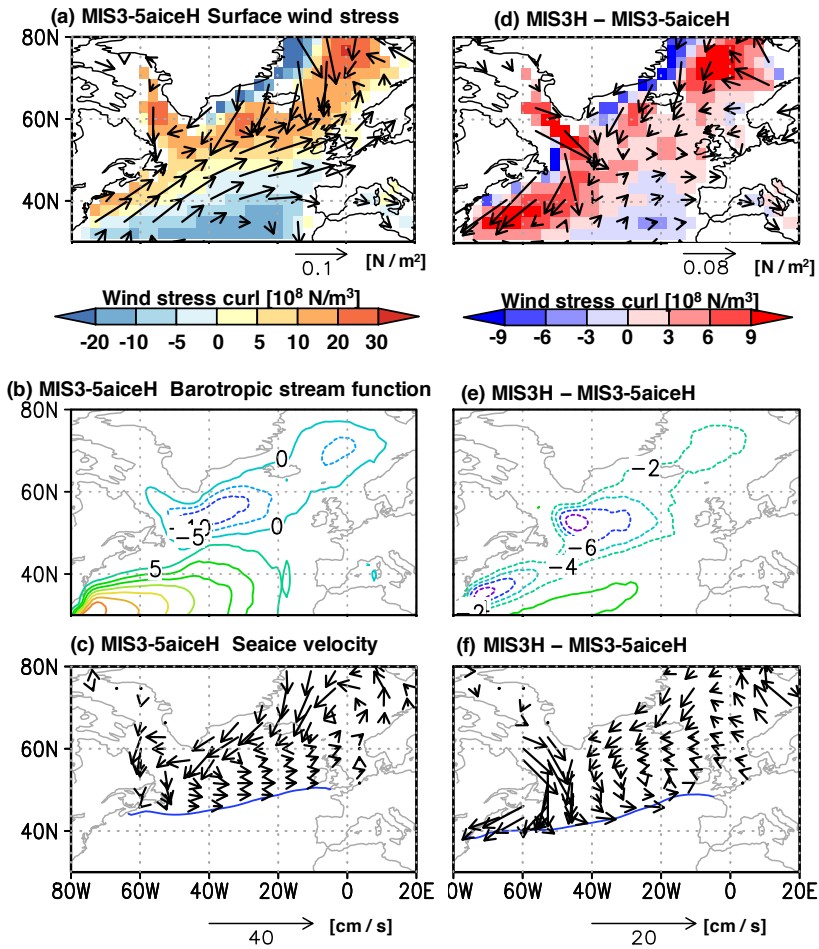

**Figure 10: Comparison of surface wind and wind-driven oceanic components between MIS3H and MIS3-5aiceH. Annual mean**
**climatology values of (a, d) surface wind stress, (b, e) barotropic streamfunction (m²/s) and (c, f) sea ice velocity (cm/s) are shown.**
**In (a), (b) and (c), results from MIS3-5aiceH are shown. In (d), (e) and (f), the differences between MIS3H and MIS3-5aiceH are**
**shown. Blue contour lines in (c) and (f) show sea ice edges (15% concentration) of MIS3-5aiceH and MIS3H, respectively. The**
**average over the last 100 years of hosing (years 401–500) is shown.**






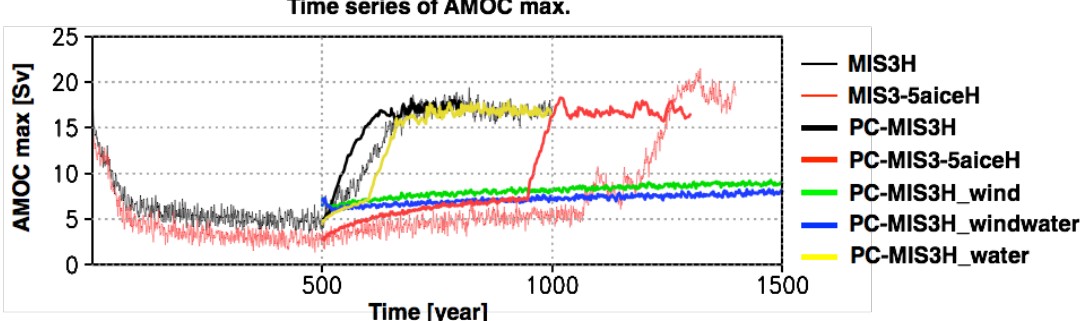

**Figure 11: Results from partially coupled experiments. Time series of the strength of the AMOC is shown. Results from the original AOGCM experiments are shown by thin lines, whereas the results from partially coupled experiments are shown in thick lines. Thin black: MIS3H, thin red: MIS3-5aiceH, thick black: PC-MIS3H, thick red: PC-MIS3-5aiceH, thick green: PC-MIS3H_wind, thick blue: PC-MIS3H_windwater, thick yellow: PC-MIS3H_water. Note that freshwater of 0.1 Sv was released to 50–70˚ N in the North Atlantic for 500 years from year 1.**


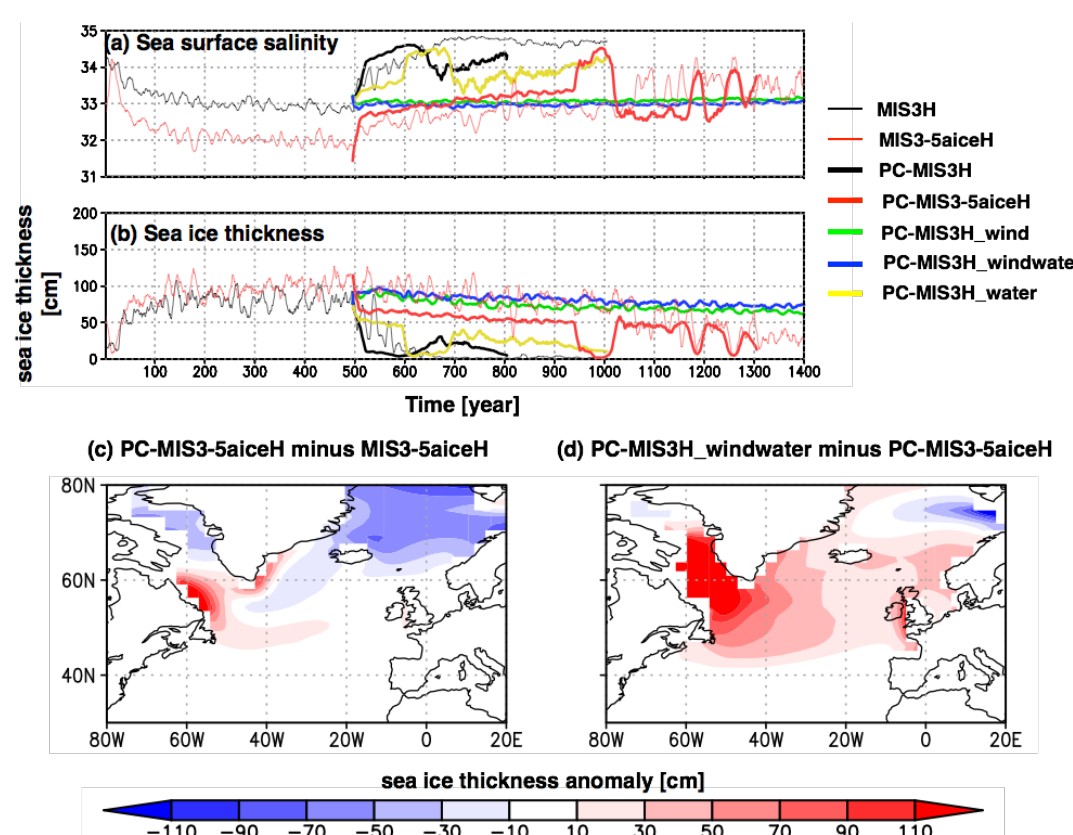

**Figure 12: Changes in sea surface conditions obtained from partially coupled experiments. Temporal evolution of February sea surface salinity (psu, a) and February sea ice thickness (cm, b) over the Irminger Sea (35–25° W, 55–63° N, left). Thin black: MIS3H, thin red: MIS3-5aiceH, thick black: PC-MIS3H, thick red: PC-MIS3-5aiceH, thick green: PC-MIS3Hwind, thick blue:**
**PC-MIS3Hwindwater, thick yellow: PC-MIS3Hwater. (c, d) Spatial maps of annual mean sea ice thickness (cm) averaged over years 501–600. Panel (c) shows the differences between a partially coupled experiment (PC-MIS3-5aiceH) and the corresponding original experiment (MIS3-5aiceH); (d) shows the effect of surface cooling by the MIS3 ice sheet (difference between PC-MIS3H_windwater and PC-MIS3-5aiceH). For (a) and (b), 11-year running means are shown.**