# Peer review of "Does a difference in ice sheets between Marine Isotope Stages 3 and 5a affect the duration of stadials?: Implications from hosing experiments"

_Climate of the Past, 2021_

## Referee Comment (RC2)

**A review of "Does a difference in ice sheets between Marine Isotope Stages 3 and 5a affect the duration of stadials?" By Sam Sherriff-Tadano et al.**

In this paper the Authors present results from a series of coupled climate model experiments to show how and why the AMOC returns to it's original state after the imposition of a large freshwater hosing. They show that the background climate that the experiments are run under causes different responses with the size of the ice sheet playing a particularly important role. Using a nice set of "partially coupled" simulations the Authors are able to show that it is a change in the winds, caused by the different ice sheet configurations, that causes the largest difference in the responses.

This study is well motivated with a clear question and the modelling simulations are well conceived. The paper is admirably succinct. However, there are some occasions where the brevity has lead to a loss of clarity. This paper ought to be published as it fits well within the scope of Climate of the Past.

I detail below some general criticisms and follow up with some more specific comments.

I would like to thank the Authors for taking the time to prepare the manuscript so well. Too often manuscripts that are closer to drafts are submitted to review. Not having to wade through poorly constructed plots with unintelligible text made reviewing this manuscript a joy rather than a chore. Thanks!

**General Comments.**

This paper presents results from a series of hosing simulations. There's a long history to these type of simulations and we can learn things about the climate system from them. However, the link between arbitrarily dumping freshwater into the North Atlantic and climate events is still not clear (Barker et al 2015). Since this paper is so clearly aimed at understanding actual climate events, DO events, there needs to be more in the Introduction about how to link the hosing simulations to real events.

Ultimately, as is stated in the Discussion, the results presented here show how the climate system responds to the cessation of an external forcing. This needs to be made clear not just at the end of the paper.

There are a number of climate models which can now simulate DO like events without the need for external forcing. It would be useful to describe these in a bit more detail in the Introduction. There are 2 reasons for this: first to show that external forcing is not the only way to change the AMOC; second, and more importantly, to give some context for how the results presented in this manuscript might apply to those simulations. For example Vetoretti and Peltier (2016/2014) describe the balance between sea ice/salinity/AMOC that is at play in their oscillations. This will clearly be modulated by the processes shown in this study. If you can link your study with that of e.g. Vetoretti and Peltier, you can make a much stronger case that the results presented here can apply far more generally than just in the case of external forced AMOC shutdowns. This Reviewer, who is a hosing sceptic, would find this much more satisfying. In the last paragraph of the Discussion this idea is mentioned I would encourage you to expand this to make the links between this study and the others clearer. Doing this should make this will make this study much more applicable to interpreting the coupled oscillations not just hosing type runs.

Figure 8 shows that the state of the climate at the end of the hosing is quite different in MIS3H and MIS3-5iceH. Could it not be the case that the different response time of the AMOC in the 2 experiments is a result of the different state from which the AMOC is recovering? The partially coupled experiments show that wind affects the response time from the MIS3 weak state, but this

does not necessarily imply that this is also the cause of the altered response time in MIS3-5aice. I think that the discussion about the winds suggests that the different sea ice and salinity distributions shown in Fig 8 can be linked to the winds but it would help a reader to be explicit about this. Fig 8 is, to this reviewer, the key figure in this paper. All of the other discussion is around trying to explain it. It would therefore help to come back to it at the end of the PC experiments to apply what you have shown.

**Specific Comments**

The title "Does a difference in ice sheets between Marine Isotope Stages 3 and 5a affect the duration of stadials?" is very snappy but ultimately in the experiments presented what determines the duration of the stadial period is how long the freshwater forcing is applied. A slightly more conservative "Does a difference in ice sheets between Marine Isotope Stages 3 and 5a affect the time it takes for the AMOC to recover from a weakening?" or similar would be a little more accurate.

Paragraphs beginning Line 246/269 – It would help to expand the description of the resumption of the AMOC in these paragraph. This would make it easier to understand the rest of the paper as a reader would better understand the set of processes (ice, salinity, convection) that lead from weak AMOC to strong. The summary sentences at the end of these paragraphs are very helpful.

Line 256 - "Four hundred years after the cessation of hosing, the surface salinity and sea ice thickness reached a quasi-equilibrium state, whereas the subsurface temperature continuously increased" how about: "an apparently steady state, however subsurface is still warming…." As it's not a quasi-eqm state.

Line 275 - "Because the surface salinity was sufficiently high in the weak phase of the AMOC, deepwater could form continuously." This suggests that deep water formation was happening during the weak AMOC phase, which I don't think is the case?

"Deepwater formation region in MIS3H" this can be seen in Fig. 5(b) correct? If so refer a reader to this figure for ease of comparison.

Line 290 - "With the southward-shifted westerly wind and strong northerly wind over the western North Atlantic, less sea ice was transported to the deepwater formation region in MIS3H" – worth saying this weakens the westerly wind formerly moving the sea ice. Confusing otherwise.

Line 291 "Therefore, even though the atmosphere was colder, less sea ice existed over the deepwater formation region." How do we know that the atmosphere was colder? You should show it.

In parts of the manuscript the link between the winds and the ice and salinity is a bit unclear. This is likely because different aspects of the overall wind change affect ice and salinity differently. So, for example, at Line 340 "It was found that the difference in surface wind played a role in causing the difference between MIS3H and MIS3-5aiceH. The cyclonic surface wind at mid-high latitudes was stronger in MIS3H than in MIS3-5aiceH. In addition, a strong northerly wind anomaly was induced over the western North Atlantic. As a result, the wind-driven transport of salt to the deepwater formation region was larger and wind-driven sea ice transport smaller in MIS3H compared with MIS3-5aiceH." It would help a reader to spell out which of the northerly anomaly and the stronger cyclonic surface wind affects sea ice and which affects salinity.

Line 318 – state that the MIS3 heat flux should lead to cooler temperatures. You say it later but a reader may already be confused.

Line 320 – "This long stadial state was caused by the very thick sea ice over the deepwater formation region, associated with stronger surface cooling by the MIS3 ice sheet (Fig 12b,d)" this is confusing, because this seems to suggest that the change in sea ice in _windwater is due to a different mechanism, surface cooling, than _wind, advection. Which is not the case? Also Fig 12b,d doesn't show stronger surface cooling in any of its plots. It would, however, be very helpful to show this.

Line 332 – "Therefore, the larger (smaller) MIS3 (MIS5a) ice sheet reduced (increased) the recovery time of the AMOC by reducing (increasing) the input of atmospheric freshwater flux over the deepwater formation region." Do not try and compress 2 sentences into 1 using brackets. It is totally unintelligible. Just write out:" Therefore, the larger MIS3 ice sheet reduced the recovery time of the AMOC by reducing the input of atmospheric freshwater flux over the deepwater formation region when compared to MIS5a."

Line 340 – add reference to Fig 10 – for a reader who comes in halfway through.

**Figures**

All time series plots need to have marks to show where the hosing is or it not occurring. E.g. Fig 8. Put some hatching over the time 0-500 to show that hosing happens here.

Fig 10. Show the deep water formation areas to allow a comparison. It's important to know where one is looking for the changes in surface fields.

---

## Author Comment (AC1)

Reply to Reviewer#1

We are grateful to reviewer#1 for his/her time in reading and commenting on our manuscript, as well as critical suggestions. As described below, we will take all of the comments raised by the reviewer into account in the revised manuscript. Below, our responses are shown in blue, and the comments by the reviewer is shown in black. Again, thank you so much for your time in reviewing our paper!

Major comments:
Line 156: How do the modeled ice sheet conditions compare to reconstructions? At least for MIS3 reconstructions exist. Gowan et al (https://www.nature.com/articles/s41467-021-21469-w).
The ice sheet configuration used in this study comes from an ice sheet model simulation of Abe-Ouchi et al. (2013, nature), which reproduces the general pattern of ice sheet evolution over the past 400 thousand years. The simulated ice sheet volume at MIS3 (36ka) is 96 m sea level equivalent, which is larger than reconstructions suggesting 40 - 90 m sea level equivalent (Grant et al. 2012 nature, Spratt and Lisieki 2016 CP, Gowan et al. 2021 Nat comm). Hence, this suggests that our study might overestimate the ice sheet impact of MIS3. Nevertheless, Gowan et al. and other studies also show a smaller ice sheet at MIS5a compared to MIS3, so the qualitative difference of ice sheet between MIS5a and MIS3 used in this study is valid. We will add a discussion on this point in the revised Discussion.

Line 160: How do you account for land-sea mask changes for the different ice sheet boundary conditions? Are they manually adjusted? Why did you choose to leave the Bering Strait open? Do you account for Bathymetric changes in your hosing experiments? Part of this is explained in Sherriff-Tadano et al., 2021 but I believe it is necessary to include some of these important aspects in the current manuscript.
The change in the land-sea mask is incorporated manually. Figure A1 shows differences in land sea mask between MIS3 and MIS5a ice sheets. Largest changes in land-sea mask locate around the Barents sea region, where new ice sheet expands in the ice sheet model. On the other hand, changes in land-sea mask near the Laurentide ice sheet and Norwegian Sea, where main convection takes place, are small. In conducting partially coupled experiment, we adjusted the location of river runoff and atmospheric freshwater flux following the changes in land sea mask by shifting it to the closest ocean grid point. We will add this explanation in the Method section.

      With respect to the Bering Strait, we did not include this change for simplicity. However, we do agree to the reviewer that the closure of Bering Strait at some point during MIS3 or MIS2, depending on sea-level reconstructions, can have an impact on the duration of stadial. We will include this discussion in the Discussion or Method in the revised version of the manuscript.

[Figure]

Fig.A1 Areas where the land-sea mask differ between MIS3 and MIS5a ice sheets.

Line 176: After reading the results I was wondering why you used monthly climatologies? The control hosing experiments and PC experiments show large differences in the timing of the recovery and I was wondering if a higher input frequency (e.g. 10-year monthly means or even

monthly means) would avoid this issue. Specifically, since your main target of exploration is the recovery time of the AMOC. How sensitive are the results to different climatologies? And is the response to different climatologies consistent?

We agree to the reviewer's concern on the choice of input frequency, given that some previous studies suggested the importance of atmospheric noise in triggering the abrupt AMOC shift (e.g. Kleppin et al. 2015, Journal of Climate). We chose to use monthly climatology in our partially coupled experiments to demonstrate the role of atmospheric forcing in a clear and simple manner. However, we also confirmed that the general result is unaffected by the choice of the input frequency. Figures A2 and A3 show a response of AMOC recovery in partially coupled experiment forced with raw daily fields from the last 100 years of hosing experiments. Generally, the experiment shows a better agreement to the original experiment (Fig. A2, reason of this is explained in the reply to the next comment). The experiments also shows the same conclusion that the surface wind effect tries to shorten the duration, while the surface cooling effect tries to increase the duration (Fig. A3). We will clarify this point in the revised manuscript.

[Figure]

Fig. A2 Time series of AMOC. Freshwater hosing of 0.1 Sv is applied during year 0 to year 500. Black and red colors correspond to MIS3 and MIS3-5aice, respectively. The original experiments are shown in thin lines, while results of partially coupled experiments forced with raw daily fields obtained from the last 100 years of the hosing experiments are shown in thick lines. This figure shows that the partially coupled experiments reproduce the original experiment better when forced with raw daily values.

[Figure]

Fig. A3 Time series of AMOC in MIS3H (black) and new partially coupled experiments after the hosing is ceased (starting from year 500). The new partially coupled experiments are forced with raw daily values from the last 100 years of the hosing experiment. Black: PC-MIS3H_daily. Blue: raw daily surface winds and atmospheric freshwater flux of the last 100 years in MIS3-5aiceH is applied to MIS3H (PC-MIS3H_windwater_daily). Yellow: Raw daily atmospheric freshwater flux of the last 100 years in MIS3-5aiceH is applied to MIS3H (PC-MIS3-5aiceH_water_daily).

Line 301: 'slightly shorter' appears to be more than 500 years in Fig. 11 for PC-MIS3-5ahice and its reference experiment. These numbers make me wonder how sensitive the experiments are to the climatology that is used. See comment to Line 176. For me the PC experiment is hardly comparable to the original experiment, also the stepwise recovery in the original experiment does not occur in the PC experiment. Also in the PC-MIS3H experiment, there is no stepwise recovery. This needs to be discussed.

As pointed out by the reviewer, the duration of staidial in PC-MIS3-5aiceH is shorter compared to MIS3-5aiceH by 200 years when comparing the onset of AMOC recovery, or shorter by 300 years when comparing the timing of fully recovered AMOC state (Fig. 11 in the original manuscript). The shorter recovery period in the partially coupled experiment is associated with the thinner sea ice over the deepwater formation (Fig. 11b and c in the original manuscript). When we use the monthly climatology, less sea ice is transported to the deepwater formation region. As a result, it gets easier for the deepwater to form and causes the early recover of the AMOC. This problem is resolved when we force the partially coupled experiment with raw daily fields as shown in Fig. A2 and in Fig. A4, which compares the sea ice thickness over Irminger Sea. We will add a discussion on this topic in the revised Discussion or Supplementary. Nevertheless, since partially coupled experiments forced with monthly climatology reproduces the general feature that MIS3-5aiceH has longer recovery time compared to MIS3H, we would like to keep using the original experiments in the revised manuscript.

      With respect to the abrupt recovery, the lack of stepwise recovery in the partially coupled experiment could be associated with weaker decadal variability and thinner sea ice over the deep water formation region. For example, in Fig. 9 of original manuscript, we have shown that the temporal cessation of deepwater formation at Irminger and Norwegian Sea associated with decadal variably could result in a temporal weakening of the AMOC. This temporal weakening of the AMOC then causes a slower recovery of the AMOC during the abrupt resumption in MIS3-5aiceH. However, we assume that this effect is weaker in partially coupled experiment since the coupling between the atmosphere and ocean is removed. While this topic is very interesting, we feel that it is beyond the scope of the study, since the main focus of the study is the duration of the recovery time of the AMOC, rather than the speed of the abrupt recovery of the AMOC. We are currently working on the interaction of decadal and millennial time-scale climate variability using partially coupled experiments and further results will be presented as a different study.

[Figure]

Fig. A4 Temporal evolution of sea ice thickness over the Irminger Sea (35W-25W, 55N-63N). Solid line corresponds to the original MIS3-5aiceH experiment, while the dashed line corresponds to a new partially coupled experiment forced with raw daily fields obtained from the last 100 years of hosing in MIS3-5aiceH. The result shows an improved reproducibility of sea ice thickness in partially coupled experiments when forced with raw daily values.

Line 385-388: What impact do uncertainties have? Previous studies have shown that uncertainties in the ice sheet reconstructions play a significant role for the glacial AMOC (e.g. Ullmann et al., 2014; www.clim-past.net/10/487/2014/). May some of these differences in the studies related to differences in the ice sheet boundary conditions? How sensitive are the results to these uncertainties? Comes also back the comment on Line 156.

Uncertainties in the volume and especially the shape of the glacial ice sheet can have a large impact on the result. For example, as we discussed in our previous paper (Sherriff-Tadano et al. 2021 CP), if the ice sheet has a thiner and wider configuration rather than thicker but smaller spatial extent, the effect of surface cooling likely gets stronger, which will favor longer staidal. Since, there's still a large debate on the volume and the shape of ice prior to LGM, we don't think we can draw a strong conclusion whether the ice sheet differences in MIS3 and MIS5a will try to reduce the duration of stadials. That is the reason why we choose to write the title and the last sentence of the abstract in a modest way. Our study, therefore, encourages further study on similar topic using other ice sheet reconstructions to better interpret the evolution of millennial-time scale climate and AMOC variability over the glacial period. We will clarify this point in the revised Discussion.

With respect to the comparison of this study and Sherriff-Tadano et al. (2021, CP), we used the same ice sheet configuration. Hence, the different sensitivity of stadial and interstadial AMOC to boundary conditions is not caused by the differences in the boundary conditions used in these studies. The result seems to be similar to ice core data, so we believe that the discussion on the different sensitivity of AMOC during stadial and interstadial is valid. But of course, as explained in the previous paragraph, if one uses different ice sheet configuration, which modifies the balance between the wind effect and surface cooling effect, different results might be obtained. One of the advantage of this study is that we clearly pointed out that the balance between the surface wind and surface cooling effects is important in determining the overall effect of ice sheets on the AMOC.

Minor comments:
All the text is written in past tense, I would suggest to write it in present. It might make it easier to distinguish between past studies and results from the present study. This would be very beneficial not only for the abstract but also the result section.

We agree to the reviewer that the present tense is better. To be honest, we wrote the first manuscript in present tense, but it was changed into past tense when we send it to an english correction service. If the past tense does not harm the clearness of the content severely, we would like to stick to the past tense at the moment, but if you disagree, we will change it to the present tense in the next round.

Line 21: I would suggest to rephrase to "under MIS5a and MIS3 boundary conditions and MIS3 boundary conditions with MIS5a ice sheets." or something similar. Otherwise it is confusing and not clear.

We will fix this.

Line 145: More than doubled is not 'slightly increased'. Please remove the word slightly.

Yes, we will remove "slightly".

Line 166: Please refer one more time to Table 1.

Yes, we will refer to table 1.

Line 176: I would recommend to remove 'that drove the ocean'. Also it should be 'a monthly climatology' or 'monthly climatologies'. Same at Line 179.

Thanks, we will fix this.

Line 180: Do you mean by noise the variability?

Yes, such as NAO and others discussed in Kleppin et al. (2015, Journal of Climate)

Line 241-242: It's not clear to me how you disentangle the effects or what you mean by: "In MIS3H, the effect of the glacial ice sheet was stronger than that of $CO_2$, and thus caused shortening of the recovery time compared with MIS5aH."
Thank you for the comment. Effects of glacial ice sheets and $CO_2$ (plus insolation) on the recovery time of AMOC can be decomposed by looking the difference between MIS3H and MIS3-5aiceH, and between MIS3-5aiceH and MIS5aH, respectively. These results show that the lower $CO_2$ causes longer recovery time whereas the larger ice sheet causes a shorter recovery time. When comparing MIS3H and MIS5aH, the duration of stadial is shorter in MIS3H despite lower $CO_2$ in this experiment. This occurs because the effect of ice sheet is stronger than that of $CO_2$. We will clarify this point in the method section of the revised manuscript.

Line 338: 'depend' needs an s.
Thanks! We will fix this.

Line 415: I was wondering whether MIROC4m can produce the afortmentioned D-O oscillations without external forcing.
Yes, we have quite a few intrinsic AMOC variably in MIROC4m. These results will be presented elsewhere.

---

## Author Comment (AC2)

Reply to Reviewer#2

We are grateful to reviewer#2 for the time in reading our manuscript, as well as critical suggestions and encouragements. As described below, we will take all of the comments raised by the reviewer into account in the revised manuscript. Below, our responses are shown in blue, and the comments by the reviewer is shown in black. Again, thank you so much for your time in reviewing our paper!

**General Comments.**
This paper presents results from a series of hosing simulations. There's a long history to these type of simulations and we can learn things about the climate system from them. However, the link between arbitrarily dumping freshwater into the North Atlantic and climate events is still not clear (Barker et al 2015). Since this paper is so clearly aimed at understanding actual climate events, DO events, there needs to be more in the Introduction about how to link the hosing simulations to real events. Ultimately, as is stated in the Discussion, the results presented here show how the climate system responds to the cessation of an external forcing. This needs to be made clear not just at the end of the paper.

We agree to the reviewer's concern that the link between the hosing experiment and actual climate events needs to be clarified in the Introduction. We will clearly state in the Introduction of the revised manuscript that

- There is a large debate on the role of freshwater hosing in DO cycles (Barker et al. 2015), and other modeling studies show a intrinsic variability that resembles DO cycles (Vettoreti and Peltier 2016, Brown and Galbraith 2016, Klockmann et al. 2018).
- We will be focusing on the situation how the climate system responds to the cessation of an external forcing through hosing experiments.
- While the cause triggering AMOC variability in hosing experiments differs from that in intrinsic oscillations, there are some similarities in the recovery process (see our reply to the next comment). Hence, there is a possibility that the outcome of the study can be applied to those obtained via intrinsic oscillations of the AMOC.

There are a number of climate models which can now simulate DO like events without the need for external forcing. It would be useful to describe these in a bit more detail in the Introduction. There are 2 reasons for this: first to show that external forcing is not the only way to change the AMOC; second, and more importantly, to give some context for how the results presented in this manuscript might apply to those simulations. For example Vetoretti and Peltier (2016/2014) describe the balance between sea ice/salinity/AMOC that is at play in their oscillations. This will clearly be modulated by the processes shown in this study. If you can link your study with that of e.g. Vetoretti and Peltier, you can make a much stronger case that the results presented here can apply far more generally than just in the case of external forced AMOC shutdowns. This Reviewer, who is a hosing sceptic, would find this much more satisfying. In the last paragraph of the Discussion this idea is mentioned. I would encourage you to expand this to make the links between this study and the others clearer. Doing this should make make this study much more applicable to interpreting the coupled oscillations not just hosing type runs.

Thank you for the encouragement! As pointed out by the reviewer, there is a similarity in the recovery process described by Vetoretti and Peltier (2016) and MIS3-5aiceH. For example, their study showed that the gradual warming at the subsurface ocean over the Irminger Sea and its balance with sea ice thickness and sea surface salinity during stadial caused the formation of deepwater. We also see a similar process operating in our hosing experiment that the gradual warming of subsurface ocean at Irminger Sea induces a deepwater formation when the sea ice is sufficiently thin and sea surface salinity is sufficiently high. We will point out this similarity in the Discussion to link our hosing studies with studies describing the mechanism of intrinsic oscillations of the AMOC.

Furthermore, as in the reply to the previous comment, we will clarify in the revised Introduction that there is a large debate on the role of freshwater hosing in DO cycles (Barker et al. 2015), and other

modeling studies show a intrinsic variability that resembles DO cycles (Vettoreti and Peltier 2016, Brown and Galbraith 2016, Klockmann et al. 2018).

Figure 8 shows that the state of the climate at the end of the hosing is quite different in MIS3H and MIS3-5iceH. Could it not be the case that the different response time of the AMOC in the 2 experiments is a result of the different state from which the AMOC is recovering? The partially coupled experiments show that wind affects the response time from the MIS3 weak state, but this does not necessarily imply that this is also the cause of the altered response time in MIS3-5aice. I think that the discussion about the winds suggests that the different sea ice and salinity distributions shown in Fig 8 can be linked to the winds but it would help a reader to be explicit about this. Fig 8 is, to this reviewer, the key figure in this paper. All of the other discussion is around trying to explain it. It would therefore help to come back to it at the end of the PC experiments to apply what you have shown.

As the reviewer pointed out, the different sea ice and salinity distributions at the end of hosing between MIS3H and MIS3-5aiceH are linked to the differences in surface winds. Ultimately, these differences in sea ice and salinity cause the different recovery time among the two. Following the reviewer's suggestion, we will explicitly explain this after the PC experiments by including more description regarding Fig. 8 in the first paragraph of the Discussion of the original manuscript.

**Specific Comments**
The title "Does a difference in ice sheets between Marine Isotope Stages 3 and 5a affect the duration of stadials?" is very snappy but ultimately in the experiments presented what determines the duration of the stadial period is how long the freshwater forcing is applied. A slightly more conservative "Does a difference in ice sheets between Marine Isotope Stages 3 and 5a affect the time it takes for the AMOC to recover from a weakening?" or similar would be a little more accurate.

We agree to the reviewer's concern that the current title is bit ambitious. On the other hand, we also feel it's quite attractive. We haven't made up our mind at this point, but at least we came up with a possible alternative title,"Does a difference in ice sheets between Marine Isotope Stages 3 and 5a affect the duration of stadials?: results from hosing experiments". We will report our plan in the revised manuscript.

Paragraphs beginning Line 246/269 – It would help to expand the description of the resumption of the AMOC in these paragraph. This would make it easier to understand the rest of the paper as a reader would better understand the set of processes (ice, salinity, convection) that lead from weak AMOC to strong. The summary sentences at the end of these paragraphs are very helpful.

We will expand the description of these two paragraphs and make the explanation of the recovery clearer in the revised manuscript.

Line 256 - "Four hundred years after the cessation of hosing, the surface salinity and sea ice thickness reached a quasi-equilibrium state, whereas the subsurface temperature continuously increased" how about: "an apparently steady state, however subsurface is still warming…." As it's not a quasi-eqm state.

Thanks for the suggestion! We will fix it as suggested.

Line 275 - "Because the surface salinity was sufficiently high in the weak phase of the AMOC, deepwater could form continuously." This suggests that deep water formation was happening during the weak AMOC phase, which I don't think is the case?
"Deepwater formation region in MIS3H" this can be seen in Fig. 5(b) correct? If so refer a reader to this figure for ease of comparison.

Some deepwater formation occurs at Irminger Sea after the cessation of hosing. This is shown in Fig. 8b, but it was not mentioned in the original manuscript, which caused some confusion. In the revised manuscript we will modify the sentence as follow to make our explanation clearer.
"Because the surface salinity was sufficiently high in the weak phase of the AMOC (Fig. 8e), deepwater could form continuously at Irminger Sea (Fig. 8b)."

Also Figure 5b shows the spatial map of convection area at the last 100 years of the hosing period. During the hosing, no deepwater formed, however, after the hosing had stopped, some convection formed over the Irminger Sea, without causing a drastic change in AMOC, but only a gradual increase in AMOC strength. We will clarify that Fig. 5b is showing the last 100 years of the hosing in the revised manuscript.

Line 290 - "With the southward-shifted westerly wind and strong northerly wind over the western North Atlantic, less sea ice was transported to the deepwater formation region in MIS3H" – worth saying this weakens the westerly wind formerly moving the sea ice. Confusing otherwise.
Following the reviewer's suggestion, we will modify the sentence as follows;
"The southward-shifted westerly wind and strong northerly wind over the western North Atlantic weaken the eastward sea ice transport to the deepwater formation region in MIS3H"

Line 291 "Therefore, even though the atmosphere was colder, less sea ice existed over the deepwater formation region." How do we know that the atmosphere was colder? You should show it.
Following the reviewer's suggestion, we will add the Fig. A5 in the revised figure. The figure, indeed, shows a colder temperature in MIS3H compared to MIS3-5aiceH.

[Figure]

Fig. A5 Annual mean surface air temperature differences between MIS3H and MIS3-5aiceH at the last 100 years of the hosing.

In parts of the manuscript the link between the winds and the ice and salinity is a bit unclear. This is likely because different aspects of the overall wind change affect ice and salinity differently. So, for example, at Line 340 "It was found that the difference in surface wind played a role in causing the difference between MIS3H and MIS3-5aiceH. The cyclonic surface wind at mid-high latitudes was stronger in MIS3H than in MIS3-5aiceH. In addition, a strong northerly wind anomaly was induced over the western North Atlantic. As a result, the wind-driven transport of salt to the deepwater formation region was larger and wind-driven sea ice transport smaller in MIS3H compared with MIS3-5aiceH." It would help a reader to spell out which of the northerly anomaly and the stronger cyclonic surface wind affects sea ice and which affects salinity.
We agree to the reviewer's point. We will clarify the relation of local surface wind and salt and sea ice transport in the revised manuscript as follows;
"It was found that the difference in surface wind played a role in causing the difference between MIS3H and MIS3-5aiceH. The cyclonic surface wind at mid-high latitudes was stronger in MIS3H than in MIS3-5aiceH. As a result, the wind-driven transport of salt to the deepwater formation was stronger in MIS3H. In addition, a strong northerly wind anomaly was induced over the western North Atlantic in MIS3H. Together with the southward shift of westerly wind, this caused a reduction of wind-driven transport of sea ice to the deepwater formation region over Irminger Sea. The higher surface salinity and thinner sea ice thickness over the deepwater formation region then increased

the probability of the recovery of the AMOC. Thus, the changes in the surface wind caused by the glacial ice sheet could contribute to a shorter stadial during MIS3 compared with MIS5."

Line 318 – state that the MIS3 heat flux should lead to cooler temperatures. You say it later but a reader may already be confused.
We will fix this following reviewer's suggestion.

Line 320 – "This long stadial state was caused by the very thick sea ice over the deepwater formation region, associated with stronger surface cooling by the MIS3 ice sheet (Fig 12b,d)" this is confusing, because this seems to suggest that the change in sea ice in _windwater is due to a different mechanism, surface cooling, than _wind, advection. Which is not the case? Also Fig 12b,d doesn't show stronger surface cooling in any of its plots. It would, however, be very helpful to show this.
Following the reviewer's comment, we will modify the sentence as follows to clarify that the same mechanism causes the increase in sea ice in both experiments;
"The long stadial states observed in these two experiments were caused by the very thick sea ice over the deepwater formation region (green and blue lines compared to red line in Fig 12b and Fig. 12d), associated with stronger surface cooling induced by the larger MIS3 ice sheet (Fig. A5)"
We will also add Fig. A5 or a similar figure to show the stronger surface cooling by the ice sheet.

Line 332 – "Therefore, the larger (smaller) MIS3 (MIS5a) ice sheet reduced (increased) the recovery time of the AMOC by reducing (increasing) the input of atmospheric freshwater flux over the deepwater formation region." Do not try and compress 2 sentences into 1 using brackets. It is totally unintelligible. Just write out:" Therefore, the larger MIS3 ice sheet reduced the recovery time of the AMOC by reducing the input of atmospheric freshwater flux over the deepwater formation region when compared to MIS5a."
Thank you for the comment! We agree it is easier to read. We will fix this following the reviewer's suggestion.

Line 340 – add reference to Fig 10 – for a reader who comes in halfway through.
We will fix this following reviewer's suggestion.

**Figures**
All time series plots need to have marks to show where the hosing is or it not occurring. E.g. Fig 8. Put some hatching over the time 0-500 to show that hosing happens here.
Thanks for the suggestion. We will fix this.

Fig 10. Show the deep water formation areas to allow a comparison. It's important to know where one is looking for the changes in surface fields.
In the last 100 years of hosing, no deepwater forms in MIS3H and MIS3-5aiceH. In fact, the figure of deepwater formation is presented in Fig. 5 d-f. However, after the cessation of hosing, some deepwater forms at Irminger Sea in both experiments before the AMOC starts to recover abruptly, which is shown in the time series figure of Fig. 8 (b,c,h,i). To make the explanation clearer, we will specify the area used for the time series analysis in Fig. 5. Hope this modification fixes the problem.

---

## Author Response (AR1)

Reply to Reviewer#1

We are grateful to reviewer#1 for the time in reading and commenting on our manuscript, as well as critical suggestions. As described below, we took all of the comments raised by the reviewer into account in the revised manuscript. Below, our responses are shown in blue, and the comments by the reviewer is shown in black. Again, thank you so much for your time in reviewing our paper!

Major comments:
Line 156: How do the modeled ice sheet conditions compare to reconstructions? At least for MIS3 reconstructions exist. Gowan et al (https://www.nature.com/articles/s41467-021-21469-w).
The ice sheet configuration used in this study comes from an ice sheet model simulation of Abe-Ouchi et al. (2013, nature), which reproduces the general pattern of ice sheet evolution over the past 400 thousand years. The simulated ice sheet volume at MIS3 (36ka) is 96 m sea level equivalent, which is larger than reconstructions suggesting 40 - 90 m sea level equivalent (Grant et al. 2012 nature, Spratt and Lisieki 2016 CP, Gowan et al. 2021 Nat comm). Hence, this suggests that our study might overestimate the ice sheet impact of MIS3. Nevertheless, Gowan et al. and other studies also show a smaller ice sheet at MIS5a compared to MIS3, so the qualitative difference of ice sheets between MIS5a and MIS3 used in this study is valid. We added the following description on this point in L166-170 of the revised manuscript;
"The volume of the MIS3 ice sheets exceeds the range of reconstructions (40- to 90-meter sea level equivalent, Grant et al. 2012, Spratt and Lisiecki 2016, Pico et al. 2017, Gowan et al. 2021), hence may cause an overestimation of the ice sheet effect. Nevertheless, the ice sheet forcing used in this study at least captures the characteristics suggested by reconstructions, which show larger ice sheets at MIS3 compared to MIS5a (Pico et al. 2017, Gowan et al. 2021)."

Line 160: How do you account for land-sea mask changes for the different ice sheet boundary conditions? Are they manually adjusted? Why did you choose to leave the Bering Strait open? Do you account for Bathymetric changes in your hosing experiments? Part of this is explained in Sherriff-Tadano et al., 2021 but I believe it is necessary to include some of these important aspects in the current manuscript.
The change in the land-sea mask is incorporated manually. Figure A1 shows differences in land sea mask between MIS3 and MIS5a ice sheets. Largest changes in land-sea mask locate around the Barents sea region, where new ice sheet expands in the ice sheet model. On the other hand, changes in land-sea mask near the Laurentide ice sheet and Norwegian Sea, where main convection takes place, are small. In conducting partially coupled experiment, we adjusted the location of river runoff and atmospheric freshwater flux following the changes in land sea mask by shifting it to the closest ocean grid point. We added the following explanation in L210-214 of the revised manuscript;
"In conducting partially coupled experiments, the location of atmospheric freshwater flux needs to be adjusted following differences in land sea mask between MIS3 and MIS5a ice sheets (Fig. S2). Largest changes appear over the Barents Sea, where new ice sheets expand. In contrast, changes in land sea mask near the Labrador Sea and Norwegian Sea, where the main oceanic convections take place (Fig. 5), are small (Fig. S2). We adjust the location of river runoff and atmospheric freshwater flux in the partially coupled experiment by shifting it to closest ocean grid points."

With respect to the Bering Strait, we did not include this change for simplicity. However, we do agree to the reviewer that the closure of Bering Strait at some point during MIS3 or MIS2, depending on sea-level reconstructions, can have an impact on the duration of stadial. We included the following discussion in L422-427 of the revised manuscript.
"we should keep in mind that there are still large uncertainties in reconstructions of the glacial ice sheets prior to LGM. For example, sea level reconstructions show a wide range of ice sheet volume from 40- to 90-meter sea level equivalent during MIS3 (Grant et al. 2012, Spratt and Lisiecki 2016, Pico et al. 2017, Gowan et al. 2021). This can directly translate into uncertainties in the quantitative effect of the ice sheets on AMOC, and also can indirectly affect the AMOC by changing the timing of the closure of Bering Strait, which may be important when interpreting DO cycles and AMOC variabilities (Hu et al. 2015)."

[Figure]

Fig.A1 Areas where the land-sea mask differ between MIS3 and MIS5a ice sheets.

Line 176: After reading the results I was wondering why you used monthly climatologies? The control hosing experiments and PC experiments show large differences in the timing of the recovery and I was wondering if a higher input frequency (e.g. 10-year monthly means or even monthly means) would avoid this issue. Specifically, since your main target of exploration is the recovery time of the AMOC. How sensitive are the results to different climatologies? And is the response to different climatologies consistent?

We agree to the reviewer's concern on the choice of input frequency, given that some previous studies suggested the importance of atmospheric noise in triggering the abrupt AMOC shift (e.g. Kleppin et al. 2015, Journal of Climate). We chose to use monthly climatology in our partially coupled experiments to demonstrate the role of atmospheric forcing in a clear and simple manner. However, we also confirmed that the general result is unaffected by the choice of the input frequency. Figures A2 and A3 show responses of AMOC recovery in partially coupled experiment forced with raw daily fields from the last 100 years of hosing experiments. Generally, the experiment shows a better agreement to the original experiment (Fig. A2, reason of this is explained in the reply to the next comment). The experiments also confirm that the same conclusion can be drawn from different input frequencies; the surface wind effect tries to shorten the duration, while the surface cooling effect tries to increase the duration (Fig. A3). We clarified this point in L190-192 of the revised manuscript and added these figures in the supplementary. "Nevertheless, similar conclusion is obtained when raw daily fields obtained from the last 100 years of the hosing period are used instead of monthly climatologies (Fig. S1)."

[Figure]

Fig. A2 Time series of AMOC. Freshwater hosing of 0.1 Sv is applied during year 0 to year 500. Black and red colors correspond to MIS3 and MIS3-5aice, respectively. The original experiments are shown in thin lines, while results of partially coupled experiments forced with raw daily fields obtained from the last 100 years of the hosing experiments are shown in thick lines. This figure

shows that the partially coupled experiments reproduce the original experiment better when forced with raw daily values.

[Figure]

Fig. A3 Time series of AMOC in MIS3H (black) and new partially coupled experiments initiated after the cessation of hosing (starting from year 500). The new partially coupled experiments are forced with raw daily values from the last 100 years of the hosing experiment. Black: PC-MIS3H_daily. Blue: raw daily surface winds and atmospheric freshwater flux of the last 100 years in MIS3-5aiceH are applied to MIS3H (PC-MIS3H_windwater_daily). Yellow: raw daily atmospheric freshwater flux of the last 100 years in MIS3-5aiceH is applied to MIS3H (PC-MIS3-5aiceH_water_daily).

Line 301: 'slightly shorter' appears to be more than 500 years in Fig. 11 for PC-MIS3-5ahice and its reference experiment. These numbers make me wonder how sensitive the experiments are to the climatology that is used. See comment to Line 176. For me the PC experiment is hardly comparable to the original experiment, also the stepwise recovery in the original experiment does not occur in the PC experiment. Also in the PC-MIS3H experiment, there is no stepwise recovery. This needs to be discussed.

As pointed out by the reviewer, the duration of staidial in PC-MIS3-5aiceH is shorter compared to MIS3-5aiceH by 200 years when comparing the onset of AMOC recovery, or shorter by 300 years when comparing the timing of fully recovered AMOC state (Fig. 11 in the original manuscript). The shorter recovery period in the partially coupled experiment is associated with the thinner sea ice over the deepwater formation (Fig. 12b and c in the original manuscript). When we use the monthly climatology, less sea ice is transported to the deepwater formation region. As a result, it gets easier for the deepwater to form and causes the early recover of the AMOC. This problem is resolved when we force the partially coupled experiment with raw daily fields as shown in Fig. A2 and in Fig. A4, the latter of which compares the sea ice thickness over Irminger Sea. We added this discussion in the revised Supplementary information. Nevertheless, since partially coupled experiments forced with monthly climatology reproduce the general feature that MIS3-5aiceH has longer recovery time compared to MIS3H, we would like to keep using the original experiments in the revised manuscript.

With respect to the abrupt recovery, the lack of stepwise recovery in the partially coupled experiment could be associated with weaker decadal variability and thinner sea ice over the deep water formation region. For example, in Fig. 9 of original manuscript, we have shown that the temporal cessation of deepwater formation at Irminger and Norwegian Sea associated with decadal variably could result in a temporal weakening of the AMOC. This temporal weakening of the AMOC then causes a slower recovery of the AMOC during the abrupt resumption in

MIS3-5aiceH. However, we assume that this effect is weaker in partially coupled experiment since the coupling between the atmosphere and ocean is removed. We add a discussion on this topic in the revised Supplementary information. While this topic is very interesting, we feel that it is beyond the scope of the study, since the main focus of the study is the duration of the recovery time of the AMOC, rather than the speed of the abrupt recovery of the AMOC. We are currently working on the interaction of decadal and millennial time-scale climate variability using partially coupled experiments, and further results will be presented as a different study.

[Figure]

Fig. A4 Temporal evolution of sea ice thickness over the Irminger Sea (35W-25W, 55N-63N). Solid line corresponds to the original MIS3-5aiceH experiment, while the dashed line corresponds to a new partially coupled experiment forced with raw daily fields obtained from the last 100 years of hosing in MIS3-5aiceH. The result shows an improved reproducibility of sea ice thickness in partially coupled experiments forced with raw daily values compared to that forced with monthly climatology.

Line 385-388: What impact do uncertainties have? Previous studies have shown that uncertainties in the ice sheet reconstructions play a significant role for the glacial AMOC (e.g. Ullmann et al., 2014; www.clim-past.net/10/487/2014/). May some of these differences in the studies related to differences in the ice sheet boundary conditions? How sensitive are the results to these uncertainties? Comes also back the comment on Line 156.

Uncertainties in the volume and especially the shape of the glacial ice sheet can have a large impact on the result. For example, as we discussed in our previous paper (Sherriff-Tadano et al. 2021 CP), if the ice sheet has a thiner and wider configuration rather than thicker but smaller spatial extent, the effect of surface cooling likely gets stronger, which will favor longer staidal. Since, there's still a large debate on the volume and the shape of ice prior to LGM, we don't think we can draw a strong conclusion whether the ice sheet differences in MIS3 and MIS5a can reduce the duration of stadials. That is the reason why we choose to write the title and the last sentence of the abstract in a modest way. Our study, therefore, encourages further study on similar topic using other ice sheet reconstructions to better interpret the evolution of millennial-time scale climate and AMOC variability over the glacial period. We clarified this point in L422-429 of the revised Discussion.

"we should keep in mind that there are still large uncertainties in reconstructions of the glacial ice sheets prior to LGM. For example, sea level reconstructions show a wide range of ice sheet volume from 40- to 90-meter sea level equivalent during MIS3 (Grant et al. 2012, Spratt and Lisiecki 2016, Pico et al. 2017, Gowan et al. 2021). This can directly translate into uncertainties in the quantitative effect of the ice sheets on AMOC, and also can indirectly affect the AMOC by changing the timing of the closure of Bering Strait, which may be important when interpreting DO cycles and AMOC variabilities (Hu et al. 2015). Furthermore, uncertainties in the shape of ice sheet may affect the balance of the surface wind and surface cooling effects on AMOC. Hence, further studies on similar topic using other ice sheet reconstructions are important to better interpret the evolution of millennial-time scale climate and AMOC variabilities over the glacial period."

With respect to the comparison of this study and Sherriff-Tadano et al. (2021, CP), we used the same ice sheet configuration. Hence, the different sensitivity of stadial and interstadial AMOC to boundary conditions is not caused by the differences in the boundary conditions used in these

studies. The result seems to be consistent with ice core data, so we believe that the discussion on the different sensitivity of AMOC during stadial and interstadial is valid. But of course, as explained in the previous paragraph, if one uses different ice sheet configuration, which modifies the balance between the wind effect and surface cooling effect, different results might be obtained. One of the advantage of this study is that we clearly pointed out that the balance between the surface wind and surface cooling effects is important in determining the overall effect of ice sheets on the AMOC. We clarified that the same ice sheet configuration is used for this study and Sherriff-Tadano et al. (2021) in the revised manuscript (L412).

Minor comments:
All the text is written in past tense, I would suggest to write it in present. It might make it easier to distinguish between past studies and results from the present study. This would be very beneficial not only for the abstract but also the result section.
We agree to the reviewer that the present tense is better. We changed it to the present tense in the revised manuscript.

Line 21: I would suggest to rephrase to "under MIS5a and MIS3 boundary conditions and MIS3 boundary conditions with MIS5a ice sheets." or something similar. Otherwise it is confusing and not clear.
Corrected (L21-L22).

Line 145: More than doubled is not 'slightly increased'. Please remove the word slightly.
We removed "slightly". (L151)

Line 166: Please refer one more time to Table 1.
Yes, we referred to table 1 in L179.

Line 176: I would recommend to remove 'that drove the ocean'. Also it should be 'a monthly climatology' or 'monthly climatologies'. Same at Line 179.
Thanks, we fixed these (L186, L189).

Line 180: Do you mean by noise the variability?
Yes, such as NAO and others discussed in Kleppin et al. (2015, Journal of Climate)

Line 241-242: It's not clear to me how you disentangle the effects or what you mean by: "In MIS3H, the effect of the glacial ice sheet was stronger than that of CO2, and thus caused shortening of the recovery time compared with MIS5aH."
Thank you for the comment. Effects of glacial ice sheets and CO2 (plus insolation) on the recovery time of AMOC can be decomposed by looking the difference between MIS3H and MIS3-5aiceH, and between MIS3-5aiceH and MIS5aH, respectively. These results show that the lower CO2 causes longer recovery time whereas the larger ice sheet causes a shorter recovery time. When comparing MIS3H and MIS5aH, the duration of stadial is shorter in MIS3H despite having lower CO2 in this experiment. This happens because the effect of ice sheet is stronger than that of CO2. We clarified this point in L258-260 of the revised manuscript.
"In MIS3H, the effect of the glacial ice sheet is strong and thus causes shortening of the recovery time compared with MIS5aH, despite having lower $CO_2$ concentration."

Line 338: 'depend' needs an s.
Thanks! We fixed this (L367).

Line 415: I was wondering whether MIROC4m can produce the afortmentioned D-O oscillations without external forcing.
Yes, we have quite a few intrinsic AMOC variabilities in MIROC4m. These results will be presented elsewhere.

Reply to Reviewer#2

We are grateful to reviewer#2 for the time in reading our manuscript, as well as critical suggestions and encouragements. As described below, we took all of the comments raised by the reviewer into account in the revised manuscript. Below, our responses are shown in blue, and the comments by the reviewer is shown in black. Again, thank you so much for your time in reviewing our paper!

**General Comments.**
This paper presents results from a series of hosing simulations. There's a long history to these type of simulations and we can learn things about the climate system from them. However, the link between arbitrarily dumping freshwater into the North Atlantic and climate events is still not clear (Barker et al 2015). Since this paper is so clearly aimed at understanding actual climate events, DO events, there needs to be more in the Introduction about how to link the hosing simulations to real events. Ultimately, as is stated in the Discussion, the results presented here show how the climate system responds to the cessation of an external forcing. This needs to be made clear not just at the end of the paper.

We agree to the reviewer's concern that the link between the hosing experiment and actual climate events needs to be clarified in the Introduction. We clarified in the revised Introduction that

- There is a debate on the role of freshwater hosing in DO cycles (Barker et al. 2015), and other modeling studies show a intrinsic variability that resembles DO cycles (Vettoreti and Peltier 2016, Brown and Galbraith 2016, Klockmann et al. 2018).
- We focus on the situation how the climate system responds to the cessation of an external forcing through hosing experiments.
- While the cause triggering AMOC variability in hosing experiments differs from that in intrinsic oscillations, there are some similarities in the recovery process (see our reply to the next comment). Hence, there is a possibility that the outcome of the study can be applied to those obtained via intrinsic oscillations of the AMOC.

These modifications can be found in L84-86

"While the timing of freshwater input and DO cycles is still debated, and that freshwater hosing may not be the cause of the AMOC weakening (Barker et al. 2015), these studies provide useful information to study DO cycles. ",
and L130-137

"We should note that, in the hosing experiments, we focus on the situation how climate system recovers from the cessation of external forcing. In contrast, recent studies with AOGCMs show intrinsic oscillations of AMOC, which resemble DO cycles, without any external forcing. For example, Vettoreti and Peltier (2016) and Sherriff-Tadano and Abe-Ouchi (2020) show in their intrinsic oscillations of AMOC that the recovery of the AMOC from weak mode to strong mode is determined by the balance among sea ice, surface salinity and subsurface ocean warming over the deepwater formation region in the North Atlantic. From the viewpoint of mechanisms, the recovery process of the AMOC in the present hosing experiments is similar to that in the intrinsic oscillations of AMOC. Therefore, our findings may not be confined to the hosing experiments or DO cycles induced by external forcing, but may also be applied to those obtained via intrinsic oscillations of the AMOC."

There are a number of climate models which can now simulate DO like events without the need for external forcing. It would be useful to describe these in a bit more detail in the Introduction. There are 2 reasons for this: first to show that external forcing is not the only way to change the AMOC; second, and more importantly, to give some context for how the results presented in this manuscript might apply to those simulations. For example Vetoretti and Peltier (2016/2014) describe the balance between sea ice/salinity/AMOC that is at play in their oscillations. This will clearly be modulated by the processes shown in this study. If you can link your study with that of e.g. Vetoretti and Peltier, you can make a much stronger case that the results presented here can apply far more generally than just in the case of external forced AMOC shutdowns. This Reviewer, who is a hosing sceptic, would find this much more satisfying. In the last paragraph of the Discussion this idea is mentioned. I would encourage you to expand this to make the links between

this study and the others clearer. Doing this should make this study much more applicable to interpreting the coupled oscillations not just hosing type runs.

Thank you for the encouragement! As pointed out by the reviewer, there is a similarity in the recovery process described by Vetoretti and Peltier (2016) and MIS3-5aiceH. For example, their study showed that the gradual warming at the subsurface ocean over the Irminger Sea and its balance with sea ice thickness and sea surface salinity during stadial caused the formation of deepwater. We also see a similar process operating in our hosing experiment that the gradual warming of subsurface ocean at Irminger Sea induces a deepwater formation when the sea ice is sufficiently thin and sea surface salinity is sufficiently high. We pointed out this similarity in L132-137, L286-288 and L449-454 of the revised manuscript to link our hosing studies with studies describing the mechanism of intrinsic oscillations of the AMOC.

Furthermore, as in the reply to the previous comment, we clarified in the revised Introduction (L84-86) that there is a debate on the role of freshwater hosing in DO cycles (Barker et al. 2015), and other modeling studies show a intrinsic variability that resembles DO cycles (Vettoreti and Peltier 2016, Brown and Galbraith 2016, Klockmann et al. 2018).

Figure 8 shows that the state of the climate at the end of the hosing is quite different in MIS3H and MIS3-5iceH. Could it not be the case that the different response time of the AMOC in the 2 experiments is a result of the different state from which the AMOC is recovering? The partially coupled experiments show that wind affects the response time from the MIS3 weak state, but this does not necessarily imply that this is also the cause of the altered response time in MIS3-5aice. I think that the discussion about the winds suggests that the different sea ice and salinity distributions shown in Fig 8 can be linked to the winds but it would help a reader to be explicit about this. Fig 8 is, to this reviewer, the key figure in this paper. All of the other discussion is around trying to explain it. It would therefore help to come back to it at the end of the PC experiments to apply what you have shown.

As the reviewer pointed out, the different sea ice and salinity distributions at the end of hosing between MIS3H and MIS3-5aiceH are linked to the differences in surface winds. Ultimately, these differences in sea ice and salinity cause the different recovery time among the two experiments. Following the reviewer's suggestion, we explicitly explained this point at the end of PC experiments by adding the following paragraph in the revised manuscript. (L359-365)
"To summarize, the shorter recovery time in MIS3H compared with MIS3-5aiceH is a result of the dominance of the surface wind effect caused by larger ice sheets. The stronger cyclonic surface winds at mid-high latitudes in MIS3H than in MIS3-5aiceH (Fig. 10d) enhance the wind-driven transport of salt to the deepwater formation in MIS3H (Fig. 10e). In addition, the strong northerly wind anomaly over the western North Atlantic and the southward shift of westerly wind cause a reduction of wind-driven transport of sea ice to the deepwater formation region over Irminger Sea in MIS3H (Fig. 10f). The higher surface salinity (Fig. 8d) and thinner sea ice thickness (Fig. 8a) over the deepwater formation region during the weak AMOC state then increase the probability of the recovery of the AMOC and cause an early recovery in MIS3H (Fig. 3). "

**Specific Comments**
The title "Does a difference in ice sheets between Marine Isotope Stages 3 and 5a affect the duration of stadials?" is very snappy but ultimately in the experiments presented what determines the duration of the stadial period is how long the freshwater forcing is applied. A slightly more conservative "Does a difference in ice sheets between Marine Isotope Stages 3 and 5a affect the time it takes for the AMOC to recover from a weakening?" or similar would be a little more accurate.

We agree to the reviewer's concern that the current title is bit ambitious. We decided to change it to the following title,"Does a difference in ice sheets between Marine Isotope Stages 3 and 5a affect the duration of stadials?: Implications from hosing experiments".

Paragraphs beginning Line 246/269 – It would help to expand the description of the resumption of the AMOC in these paragraph. This would make it easier to understand the rest of the paper as a

reader would better understand the set of processes (ice, salinity, convection) that lead from weak AMOC to strong. The summary sentences at the end of these paragraphs are very helpful.

Following the reviewer's comment, we clarified the explanations of these two paragraphs by increasing the explanation on the relation among AMOC strength, sea surface salinity, sea ice thickness and deepwater formation. Additionally, we referred to the figure at each sentence to make it easier for readers to relate the explanation and the figure. Hope this modifications improved our description (L263-302).

Line 256 - "Four hundred years after the cessation of hosing, the surface salinity and sea ice thickness reached a quasi-equilibrium state, whereas the subsurface temperature continuously increased" how about: "an apparently steady state, however subsurface is still warming…." As it's not a quasi-eqm state.

Thanks for the suggestion! We fixed it (L273-L274).

Line 275 - "Because the surface salinity was sufficiently high in the weak phase of the AMOC, deepwater could form continuously." This suggests that deep water formation was happening during the weak AMOC phase, which I don't think is the case?
"Deepwater formation region in MIS3H" this can be seen in Fig. 5(b) correct? If so refer a reader to this figure for ease of comparison.

Some deepwater formation occurs at Irminger Sea after the cessation of hosing. This is shown in Fig. 8b, but it was not mentioned in the original manuscript, which caused some confusion. In the revised manuscript we modified the sentence as follows to make our explanation clearer (L290-296).
"At first, during the hosing period, sea surface salinity is higher and sea ice thickness is thinner compared with MIS3-5aiceH (Fig. 8a, d, e), which are favourable conditions to induce deepwater formation. Note that at this point, no deepwater forms at northern North Atlantic (Fig. 5d and Fig. 8b, h). After the cessation of freshwater hosing, however, the initial increase of surface salinity triggers a deepwater formation over Irminger Sea (Fig. 8b, e). Because the surface salinity is already sufficiently high in the last 100 years of the hosing period (Fig. 8e), deepwater can form continuously over Irminger Sea (Fig. 8b). As a result, vertical mixing occurs continuously and further increases surface salinity and decreases sea ice thickness over the Irminger Sea and Greenland Sea. "

Also Figure 5b shows the spatial map of convection area before the hosing, and Figure 5d shows the similar figure at the last 100 years of the hosing period. During the hosing, no deepwater forms, however, after the cessation of hosing, some convections initiate over the Irminger Sea, without causing a drastic change in AMOC, but only a gradual increase in AMOC strength. We clarified the explanation of Figure caption (L731-732)
"In (a-c), deepwater formation regions before the hosing are displayed, while in (d-f), deepwater formation regions of the last 100 years of the hosing period are shown. "

Line 290 - "With the southward-shifted westerly wind and strong northerly wind over the western North Atlantic, less sea ice was transported to the deepwater formation region in MIS3H" – worth saying this weakens the westerly wind formerly moving the sea ice. Confusing otherwise.

Following the reviewer's suggestion, we modified the sentence as follows (L310-L312);
"The southward-shifted westerly wind and strong northerly wind over the western North Atlantic act to reduce the eastward transport of sea ice to the deepwater formation region in MIS3H (Fig. 10c, f)."

Line 291 "Therefore, even though the atmosphere was colder, less sea ice existed over the deepwater formation region." How do we know that the atmosphere was colder? You should show it.

Following the reviewer's suggestion, we added Fig. A5 in the Supplementary. The figure, indeed, shows a colder temperature in MIS3H compared to MIS3-5aiceH.

[Figure]

Fig. A5 Annual mean surface air temperature differences between MIS3H and MIS3-5aiceH at the last 100 years of the hosing.

In parts of the manuscript the link between the winds and the ice and salinity is a bit unclear. This is likely because different aspects of the overall wind change affect ice and salinity differently. So, for example, at Line 340 "It was found that the difference in surface wind played a role in causing the difference between MIS3H and MIS3-5aiceH. The cyclonic surface wind at mid-high latitudes was stronger in MIS3H than in MIS3-5aiceH. In addition, a strong northerly wind anomaly was induced over the western North Atlantic. As a result, the wind-driven transport of salt to the deepwater formation region was larger and wind-driven sea ice transport smaller in MIS3H compared with MIS3-5aiceH." It would help a reader to spell out which of the northerly anomaly and the stronger cyclonic surface wind affects sea ice and which affects salinity.

We agree to the reviewer's point. We clarified the relation of local surface wind and salt and sea ice transport in the revised manuscript as follows (L360-L365);
"The stronger cyclonic surface winds at mid-high latitudes in MIS3H than in MIS3-5aiceH (Fig. 10d) enhance the wind-driven transport of salt to the deepwater formation in MIS3H (Fig. 10e). In addition, the strong northerly wind anomaly over the western North Atlantic and the southward shift of westerly wind cause a reduction of wind-driven transport of sea ice to the deepwater formation region over Irminger Sea in MIS3H (Fig. 10f). The higher surface salinity (Fig. 8d) and thinner sea ice thickness (Fig. 8a) over the deepwater formation region during the weak AMOC state then increase the probability of the recovery of the AMOC and cause an early recovery in MIS3H (Fig. 3). "

Line 318 – state that the MIS3 heat flux should lead to cooler temperatures. You say it later but a reader may already be confused.
We fixed it following reviewer's suggestion (L339).

Line 320 – "This long stadial state was caused by the very thick sea ice over the deepwater formation region, associated with stronger surface cooling by the MIS3 ice sheet (Fig 12b,d)" this is confusing, because this seems to suggest that the change in sea ice in _windwater is due to a different mechanism, surface cooling, than _wind, advection. Which is not the case? Also Fig 12b,d doesn't show stronger surface cooling in any of its plots. It would, however, be very helpful to show this.

Following the reviewer's comment, we modified the sentence as follows to clarify that the same mechanism causes the increase in sea ice in both experiments (L338-L343);
"the AMOC does not recover in PC-MIS3H_wind during the integration, despite having the same surface wind forcing as in PC-MIS3-5aiceH, which recovers around year 900. A similar feature is also observed in PC-MIS3H_windwater (Table 2), where the model is forced with the surface cooling of the MIS3 ice sheet (MIS3H) and the surface wind and atmospheric freshwater flux of the

MIS5a ice sheet (MIS3-5aiceH). The long stadial states observed in these two experiments are caused by the very thick sea ice over the deepwater formation region (green and blue lines compared to the red line in Fig. 12b, see also Fig. 12d), associated with stronger surface cooling by the MIS3 ice sheet (Fig. S5) "
We also added Fig. A5 in Supplementary to show the stronger surface cooling by the ice sheet.

Line 332 – "Therefore, the larger (smaller) MIS3 (MIS5a) ice sheet reduced (increased) the recovery time of the AMOC by reducing (increasing) the input of atmospheric freshwater flux over the deepwater formation region." Do not try and compress 2 sentences into 1 using brackets. It is totally unintelligible. Just write out:" Therefore, the larger MIS3 ice sheet reduced the recovery time of the AMOC by reducing the input of atmospheric freshwater flux over the deepwater formation region when compared to MIS5a."
Thank you for the comment! We agree it is easier to read. We fixed it following the reviewer's suggestion (L354-356).
"Therefore, the larger MIS3 ice sheet reduces the recovery time of the AMOC by reducing the input of atmospheric freshwater flux over the deepwater formation region when compared to MIS5a ice sheet."

Line 340 – add reference to Fig 10 – for a reader who comes in halfway through.
We fixed this following reviewer's suggestion (L370).

**Figures**
All time series plots need to have marks to show where the hosing is or it not occurring. E.g. Fig 8. Put some hatching over the time 0-500 to show that hosing happens here.
Thanks for the suggestion. We fixed this in the revised figures 8, 11, 12.

Fig 10. Show the deep water formation areas to allow a comparison. It's important to know where one is looking for the changes in surface fields.
In the last 100 years of hosing, no deepwater forms in MIS3H and MIS3-5aiceH. In fact, the figures of deepwater formation are presented in Fig. 5 d-f. However, after the cessation of hosing, some deepwater forms at Irminger Sea in both experiments before the AMOC starts to recover abruptly, which is shown in the time series figure of Fig. 8 (b,c,h,i). To make the explanation clearer, we clarified the area used for the time series analysis in Fig. 5. Hope this modification fixes the problem.

---

## Author Response (AR2)

Dear Dr. Menviel

Thank you very much for your time in evaluating our manuscript as well as the comments which helped to improve our manuscript.
We've fixed the error you kindly pointed out in the updated manuscript and uploaded the data of MIROC simulation results.

Best wishes
Sam